# Flexible computation of object motion and depth based on viewing geometry inferred from optic flow

Zhe-Xin Xu [1,2] ✉, Jiayi Pang[1,3], Akiyuki Anzai[1] & Gregory C. DeAngelis [1]

We move our eyes and head to sample the visual environment. While these movements are essential for survival, they greatly complicate the analysis of retinal image motion. Our brain must account for the visual consequences of self-motion to perceive the 3D layout and motion of objects in a scene. We show that traditional models of visual compensation for eye movements fail when the eye both translates and rotates, and we propose a theory that computes both motion and depth in more natural viewing geometries. Consistent with our theoretical predictions, humans exhibit distinct perceptual biases when different viewing geometries are simulated by optic flow, and these biases occur without training or feedback. A neural network model trained to perform the same tasks suggests that viewing geometry modulates the joint tuning of neurons for retinal and eye velocity to mediate these adaptive computations. Our findings unify previously separate bodies of work by demonstrating that the brain adaptively perceives the dynamic 3D environment according to viewing geometry inferred from optic flow.

From hawks catching prey to tennis players hitting a topspin forehand, humans and other animals frequently move their bodies to interact with the world. This requires processing sensory signals that arise from changes in the environment (e.g., objects moving in the world), as well as sensory signals that arise from our own actions (e.g., self-motion). A key challenge arises in these computations: the brain needs to decompose sensory signals into contributions caused by events in the environment and those resulting from one's own actions[1]. This type of computation is an example of causal inference[2–4]. To identify components of visual input that arise during self-motion, one must infer their own viewing geometry, namely, how the eyes translate and rotate relative to the scene as a result of eye, head, or body movements. As we will demonstrate, correctly computing the motion and 3D location of objects depends crucially upon correctly inferring one's viewing geometry.

A classic example of how the brain compensates for visual consequences of action involves smooth pursuit eye movements, which we use to track objects of interest[5,6]. Many studies have examined how the brain compensates for the visual consequences of pursuit eye movements, and how this affects visual perception[5–24]. For example, the Filehne illusion and Aubert-Fleischl phenomenon occur when the visual signal caused by a smooth eye movement is not accurately compensated[7–9]. Theories that have been proposed to account for these perceptual phenomena[10–12,15,24,25] generally share a common computational motif in which the brain compensates for the visual consequences of smooth pursuit by performing a vector subtraction of a reference signal that is related to eye velocity[12,15,22] (Fig. 1). While these models can account for biases in visual perception produced by a pure eye rotation, we demonstrate that they fail dramatically for even simple combinations of eye translation and rotation. Since the majority of previous empirical studies only tested visual stimuli on a 2D display, these limitations of the classic vector subtraction model appear to be largely unappreciated. In the present study, we show that computing object motion and depth in the world requires more than a simple vector subtraction, and that the brain flexibly interprets specific components of retinal image motion based on the 3D viewing geometry inferred from optic flow.

[1]Department of Brain and Cognitive Sciences, Center for Visual Science, University of Rochester, Rochester, NY, USA. [2]Department of Neurobiology, Harvard Medical School, Boston, MA, USA. [3]Department of Cognitive and Psychological Sciences, Brown University, Providence, RI, USA. ✉ e-mail: brian_xu@hms.harvard.edu

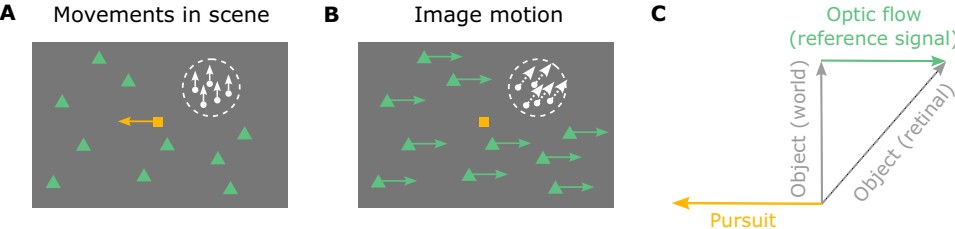

**Fig. 1 | Schematic illustration of smooth pursuit eye movement and its visual consequences in a 2D scene. A** This diagram illustrates movements in the scene, including a pursuit target (yellow square) moving leftwards and an object (white patch of random dots) moving upwards. Green triangles depict stationary background elements in the scene. **B** The resulting image motion for the scenario of (**A**), shown in screen coordinates, assuming that the observer accurately pursues the yellow target. The image motion of the green triangles reflects optic flow generated by the eye movement (green arrows), and the image motion of the white object (white arrows) reflects both its motion in the world and the observer's eye movement. **C** The object's motion in the world (solid gray arrow) can be obtained by subtracting the optic flow vector (green arrow) from the retinal image motion of the object (dashed gray arrow), which is equivalent to adding pursuit eye velocity (yellow arrow) to retinal image motion.

To illustrate the importance of 3D viewing geometry, consider how the effects of eye movement on visual input depend on scene structure. A typical paradigm for studying the effect of smooth pursuit on visual perception is shown in Fig. 1A. A fixation target (yellow square) moves across the center of the screen, while a visual stimulus (e.g., a random-dot patch) appears at a particular location on the screen and moves independently (Fig. 1A, white dots and arrows). The observer tracks the fixation target by making a leftward smooth pursuit eye movement, which results in rightward optic flow (Fig. 1B, green arrows). For the moving object (white dots), retinal image motion reflects both its motion in the scene (world coordinates) and the optic flow resulting from eye movement. To compute its motion in the world, the observer needs to subtract the optic flow vector from the retinal image motion (or, equivalently, add eye velocity to it; Fig. 1C). This scenario typically occurs when the observer remains stationary, and only the eyes rotate (Fig. 2A). The object's distance, or depth, does not affect the computation of its motion in world coordinates because the rotational flow field that results from a pure eye rotation is depth-invariant[26] (Fig. 2A). We hereby refer to this viewing geometry as Pure Rotation (R). See Supplementary Movie 1 for a demonstration of a pure rotational flow field.

Now consider a simple extension of the viewing geometry in which the observer translates laterally while maintaining visual fixation on a fixed point in the scene by counter-rotating their eyes (Fig. 2B). We refer to this viewing geometry as Rotation + Translation (R+T). By introducing this lateral translation, the same leftward pursuit eye movement becomes associated with a drastically different optic flow pattern (Fig. 2B, green arrows). In this geometry, the scene rotates around the point of fixation, and motion parallax (MP) cues for depth become available. Stationary objects at different depths relative to the fixation point move with different velocities on the retina, and the retinal image speed increases with distance from the fixation point (Fig. 2B; see Supplementary Movie 2 for a demonstration). Stationary elements in the scene nearer than the fixation point move in the same direction as the eye, while far elements move in the opposite direction (Fig. 2B, green triangles and arrows). Because there are a variety of optic flow vectors associated with the same eye movement in the R+T geometry, compensating for the visual consequences of pursuit can no longer be a simple vector subtraction; one must consider the depth of objects.

Therefore, the visual consequences of a smooth pursuit eye movement differ greatly depending on viewing geometry, and it is crucial to understand that different computations are typically performed to interpret the scene in these two viewing geometries: coordinate transformation in the R geometry and estimation of depth from motion parallax in the R+T geometry (Fig. 2C, D). Coordinate transformation (CT) refers to transforming object motion from retina-centered coordinates to world-centered coordinates[15,27–29] (Fig. 2C). In

## Viewing Geometries

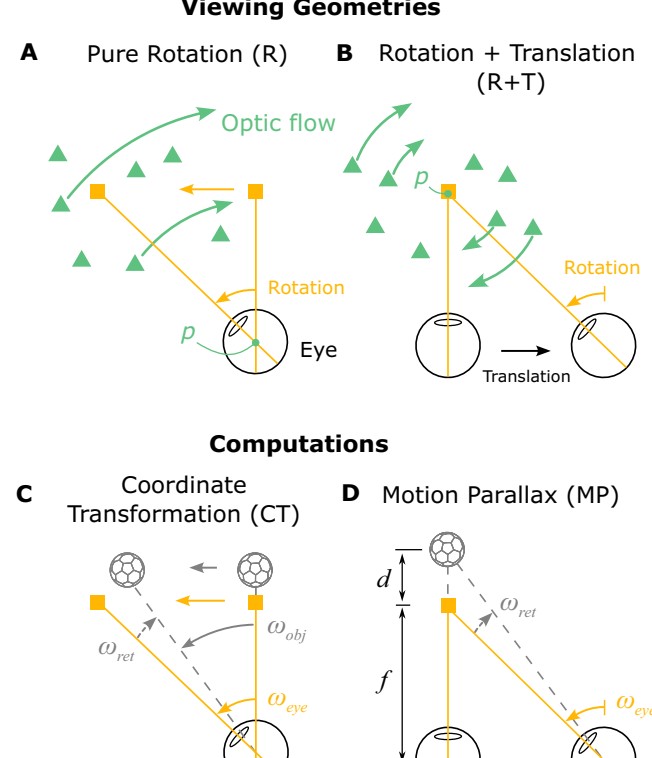

**Fig. 2 | Schematic illustration of two viewing geometries and corresponding computations that can be performed. A** Top-down view of the Pure Rotation (R) viewing geometry, in which a stationary observer rotates their eye to track a moving fixation target (yellow square), resulting in an optic flow field (green arrows, shown for a subset of triangles for clarity) that rotates around the eye (rotation pivot, *p*) in the opposite direction of eye movement. **B** In the Rotation + Translation (R+T) viewing geometry, the observer translates laterally and counter-rotates their eye to maintain fixation on a stationary target (yellow square), producing optic flow vectors (green arrows) in opposite directions for near and far objects (green triangles). This optic flow pattern is effectively a rotational flow field around the fixation target (rotation pivot, *p*). **C** In the Pure Rotation (R) viewing geometry, the retinal image motion of a moving object (soccer ball shape), $\omega_{ret}$ (dashed gray arrow), reflects both its motion in the world, $\omega_{obj}$ (solid gray arrow), and the velocity of eye rotation, $\omega_{eye}$ (yellow arrow). By taking a vector sum between $\omega_{ret}$ and $\omega_{eye}$, the velocity of the object can be transformed from retinal coordinates to world coordinates, hereafter referred to as a coordinate transformation (CT). **D** In the Rotation + Translation (R+T) viewing geometry, the retinal image motion of a stationary object (soccer ball shape), $\omega_{ret}$ (dashed gray arrow), depends on where the object is located in depth, *d*, and the rotational eye velocity, $\omega_{eye}$ (yellow arrow). By computing the ratio between $\omega_{ret}$ and $\omega_{eye}$, the depth of the object can be obtained from the motion parallax (MP) cue.

the R geometry, since it is unlikely that the entire scene rotates around the eye due to external causes, it is natural for the brain to attribute rotational optic flow to eye rotation and to represent object motion relative to the head by subtracting optic flow from the retinal image. For example, leftward smooth pursuit would induce rightward optic flow and a horizontal (leftward) bias in perceived direction of a moving object relative to its image motion, as observed empirically[19,30,31]. On the other hand, in the R+T geometry, motion parallax (MP) provides valuable information about the depth of stationary objects[32–36]. Specifically, depth can be computed as the ratio of its retinal image motion and the pursuit eye velocity[35] (Fig. 2D). When the eye translates and counter-rotates horizontally, as illustrated in Fig. 2B, the horizontal component of an object's motion could be attributed to depth, especially if other depth cues are not in conflict. Therefore, in the R+T geometry, it is natural for the brain to attribute at least some of the horizontal component of motion to depth while the vertical component is attributed to independent object motion, thus leading to a vertical bias in perceived direction of the object. Thus, as we formalize below, accounting for the visual consequences of eye movements under different viewing geometries predicts systematic patterns of biases in motion and depth perception that otherwise may not be anticipated.

CT computations and estimation of depth from MP have been studied extensively in terms of behavior[9,12,15,25,28,29,33,34,37–46] and neural mechanisms[39,47–57]. However, previous studies generally treat these two phenomena as separate and unrelated. Interestingly, in these two viewing geometries, the same two signals—retinal velocity and eye velocity—are typically combined in different ways: summation to compute object motion in the world (CT, Eq. 3), and division to estimate depth based on MP (Eq. 4). This raises an important question: how does the brain infer the relevant viewing geometry and use this information to compute object motion and depth in a context-dependent fashion?

We demonstrate that the R and R+T viewing geometries are specific instances of a general framework that explains interactions between motion and depth perception under a range of self-motion conditions. While the computations for object motion and depth take apparently distinct forms, involving addition and division respectively, they can be unified under the same framework when considering viewing geometry. We conduct psychophysical experiments to characterize the effects of viewing geometry, simulated by optic flow, on the perception of object motion and depth. Our findings show that humans flexibly and automatically (without any training or feedback) compute motion and depth based on the simulated viewing geometry, even in the absence of extra-retinal signals about eye movement. Rather than being a nuisance variable to suppress, visual image motion induced by smooth eye movements provides a powerful input for flexibly computing object motion and depth in a context-specific manner.

To investigate potential neural substrates of these flexible computations of motion and depth, we train recurrent neural networks to perform these tasks and compare the representations learned by hidden units of the network with those in the primate visual cortex. We show that task-optimized recurrent neural networks exhibit adaptive representations roughly similar to those found in neurons in the macaque middle temporal (MT) area. Our work thus reveals the computational principles and a potential neural basis of how we perceive the 3D world while in motion.

## Results

We present a unifying theory that links computations of object motion and depth with viewing geometry by considering the optic flow patterns produced by eye rotation and translation. Our theory predicts striking differences in motion and depth perception between the R and R+T viewing geometries, which we validate by performing psychophysical experiments with human participants. We demonstrate that humans automatically and flexibly alter their estimation of motion and depth based on the viewing geometry simulated by optic flow. Finally, we show that recurrent neural networks trained to compute object motion and depth exhibit non-separable retinal and eye velocity tuning similar to neurons found in area MT, suggesting a potential neural basis for computing motion and depth in a viewing geometry–dependent manner.

### Different viewing geometries generate distinct optic flow fields

We start by asking which cues are pivotal in shaping beliefs about viewing geometry. While the retinal image motion of the object and eye-in-head rotation may be identical between the R and R+T viewing geometries (Fig. 2C, D), the optic flow field generated by eye movements clearly differentiates the viewing geometry (Fig. 2A, B and Supplementary Movies 1 and 2). In the R geometry, the observer's head is stationary, and only the eyes rotate; therefore, the optic flow field reflects rotation of the scene around the eye (Fig. 2A, point $p$). In the R+T geometry, the observer's head translates laterally, and the eyes counter-rotate to maintain fixation on a world-fixed point. Therefore, the optic flow field reflects rotation of the scene around the fixation point (Fig. 2B, point $p$).

In essence, the main difference between R and R+T viewing geometries can be captured by a single parameter: the rotation pivot of the optic flow field produced by eye movements. To formalize the relationship between retinal image motion, eye rotation, object motion, and depth, we consider a more general viewing geometry depicted in Fig. 3; this generalization encompasses the R and R+T geometries. The retinal image motion of an object, $\omega_{\text{ret}}$, is a combination of the object's scene-relative motion, $\omega_{\text{obj}}$, motion parallax from the observer's translation (which depends on depth), $\omega_{\text{ret}}^{T}$, and optic flow produced by the observer's pursuit eye movement, $\omega_{\text{ret}}^{P}$ (Fig. 3B):

$$\omega_{\text{ret}} = \omega_{\text{obj}} + \omega_{\text{ret}}^{T} + \omega_{\text{ret}}^{P} \tag{1}$$

Therefore, object motion in world coordinates can be expressed as (see Supplementary Information for derivation):

$$\omega_{\text{obj}} = \omega_{\text{ret}} + (1 - (1 + d')p')\omega_{\text{eye}} \tag{2}$$

Here, $\omega_{\text{obj}}$, $\omega_{\text{ret}}$, and $\omega_{\text{eye}}$ denote the angular velocities of object motion in world coordinates, its retinal motion, and eye rotation. $d'$ represents the object's depth, $d$, normalized by viewing distance, $f$: $d' \triangleq d/f$. When $d' = 0$, the object is at the same depth as the fixation plane, whereas $d' < 0$ means near and $d' > 0$ means far compared to the fixation plane. Similarly, $p'$ represents the normalized rotation pivot, $p' \triangleq p/f$, where $p$ is the distance from the rotation pivot to the cyclopean eye. Therefore, $p' = 0$ corresponds to the R geometry and $p' = 1$ indicates the R+T geometry.

When $p' = 0$ (R geometry), object motion in world coordinates is the sum of retinal and eye velocities, thus capturing the coordinate transformation computation:

$$\omega_{\text{obj}} = \omega_{\text{ret}} + \omega_{\text{eye}}. \tag{3}$$

When $p' = 1$ (R+T geometry) and the object is stationary in the world, $\omega_{\text{obj}} = 0$, the object's relative depth is the ratio between retinal and eye velocities, resulting in the approximate form of the motion-pursuit law[35]:

$$d' \triangleq \frac{d}{f} = \frac{\omega_{\text{ret}}}{\omega_{\text{eye}}}. \tag{4}$$

Our derivation of Eq. 2 (see Supplementary Information for details) thus provides a general framework that includes the R and R+T

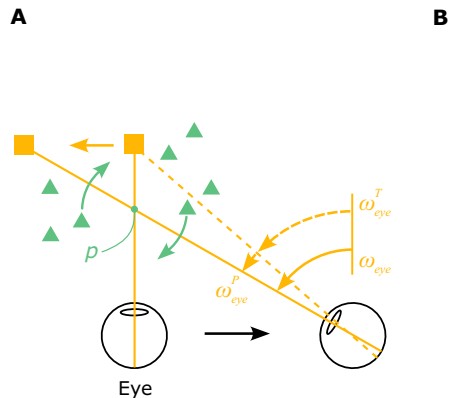

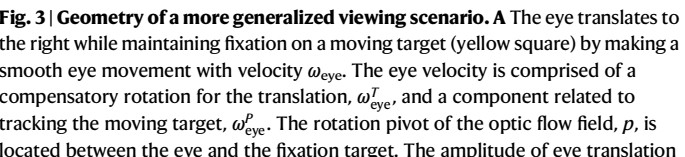

**Fig. 3 | Geometry of a more generalized viewing scenario. A** The eye translates to the right while maintaining fixation on a moving target (yellow square) by making a smooth eye movement with velocity $\omega_{eye}$. The eye velocity is comprised of a compensatory rotation for the translation, $\omega^T_{eye}$, and a component related to tracking the moving target, $\omega^P_{eye}$. The rotation pivot of the optic flow field, $p$, is located between the eye and the fixation target. The amplitude of eye translation and rotation is exaggerated for the purpose of illustration. **B** When an object (soccer ball shape) is located at depth, $d$, and moves independently in the world, its retinal image velocity, $\omega_{ret}$, is determined by its own motion in the world, $\omega_{obj}$, motion parallax produced by the observer's translation, $\omega^T_{ret}$, and image motion resulting from the pursuit eye movement, $\omega^P_{ret}$. $\omega_{eye}$, $\omega^T_{eye}$, $\omega^P_{eye}$, and $\omega_{obj}$ are defined relative to a fixed point in the world, and $\omega_{ret}$, $\omega^T_{ret}$, and $\omega^P_{ret}$ are relative to the eye.

viewing geometries, and thereby links together the computations of object motion and depth for a moving observer. While the addition and division computations for computing motion and depth appear quite different on the surface, both operations can be expressed as a single computation when we incorporate the rotation pivot of optic flow. Thus, it suggests that the brain transitions between these operations when optic flow implies different viewing geometries.

This finding raises important questions. Do people use optic flow to infer their viewing geometry? Do they perceive object motion and depth differently when viewing optic flow that simulates different viewing geometries? We conducted a series of psychophysical experiments to measure motion and depth perception in humans while presenting optic flow patterns simulating different viewing geometries. We also examined whether effects are substantially different when participants pursue a target with their eyes, as compared to when pursuit is visually simulated.

## Humans rely on 3D viewing geometry to infer motion in the world

In Experiment 1, human participants performed a motion estimation task. Specifically, we presented an object moving with a fixed set of directions on the retina while simulating different viewing geometries with large-field background motion (Fig. 4; see "Methods" for details). The object and optic flow were presented for 1 s, after which a probe stimulus appeared, and participants adjusted a dial to match the motion direction of the probe with that of the object (Fig. 6A, B).

Four main experimental conditions were interleaved: two eye-movement conditions times two simulated viewing geometries (Fig. 4 and Supplementary Movies 3–6). The two eye-movement conditions include: (1) the Pursuit condition, in which participants tracked a moving target by making smooth pursuit eye movements while the head remained stationary (Fig. 4A, C), and (2) the Fixation condition, in which participants fixated on a stationary target at the center of the screen, and eye movements were simulated by background motion (Fig. 4B, D). The two background conditions include: (1) the R condition, in which background dots simulated the R viewing geometry (Fig. 4A, B), and (2) the R+T condition, in which the background simulated the R+T geometry (Fig. 4C, D). Simulated eye translation in the R+T condition was always horizontal (i.e., along the interaural axis), and target motion in the Pursuit condition was also always horizontal. Thus, all real and simulated pursuit eye movements were horizontal (leftward or rightward). Notably, participants did not receive any form of feedback during performance of the task. In addition to the four

main conditions, a control condition, in which no background dots were present and thus no cues for viewing geometry were available, was interleaved to measure baseline motion estimation performance (Supplementary Movies 7–8).

Because the eye translation and/or rotation simulated by background motion was always horizontal, we expect the two viewing geometries, R and R+T, to affect perception of the object's horizontal motion component. For the R viewing geometry, in which the brain naturally performs CT, horizontal eye velocity should be added to the object's retinal motion (Eq. 3), resulting in a perceptual bias towards the horizontal (Fig. 5, top row, orange arrows; Fig. 6C, orange curves). Because the object was presented monocularly and its size was kept constant on the screen across conditions, the object's depth should be ambiguous in the R geometry (Fig. 5, top row, purple; Fig. 7C, orange band), for which optic flow is depth-invariant. Conversely, in the R+T case, we expect eye velocity to be combined with the object's horizontal retinal motion to compute depth from MP based on the motion-pursuit law[35] (Eq. 4). Because of the absence of other depth cues, we hypothesize that the horizontal component of the object's retinal motion will be explained away as MP resulting from observer translation (Fig. 5, bottom row, purple arrows). Consequently, only the remaining vertical motion component will be perceived as object motion (Fig. 5, bottom row, blue arrows). In this case, we expect participants to show a perceptual bias toward vertical directions (Fig. 5, bottom row, blue arrows; Fig. 6C, blue curves).

Observers may not accurately infer the viewing geometry from optic flow. In the R geometry, they might underestimate their eye velocity; in the R+T geometry, they might only attribute a portion of the horizontal velocity component to depth. Therefore, our theoretical framework incorporates two discount factors into our predictions (Eq. 9; Fig. 6C), which are conceptually analogous to a flow parsing gain[40,58,59].

Indeed, we found systematic biases in motion direction reports for all participants, and the pattern of biases is generally consistent with our predictions (compare Fig. 6D, E and Supplementary Fig. S3 with Fig. 6C). Specifically, we found an overall bias toward reporting horizontal directions in the R viewing geometry for both simulated (Fixation) and real (Pursuit) eye rotation conditions (Fig. 6D, E and Supplementary Fig. S3, orange). In the R+T viewing geometry, we found an overall bias toward reporting vertical directions in the Fixation condition, with mixed results for the Pursuit condition as described further below (Fig. 6D, E and Supplementary Fig. S3, blue). As a control, we found no bias in the Fixation condition when there were no background dots, indicating that participants accurately perceived

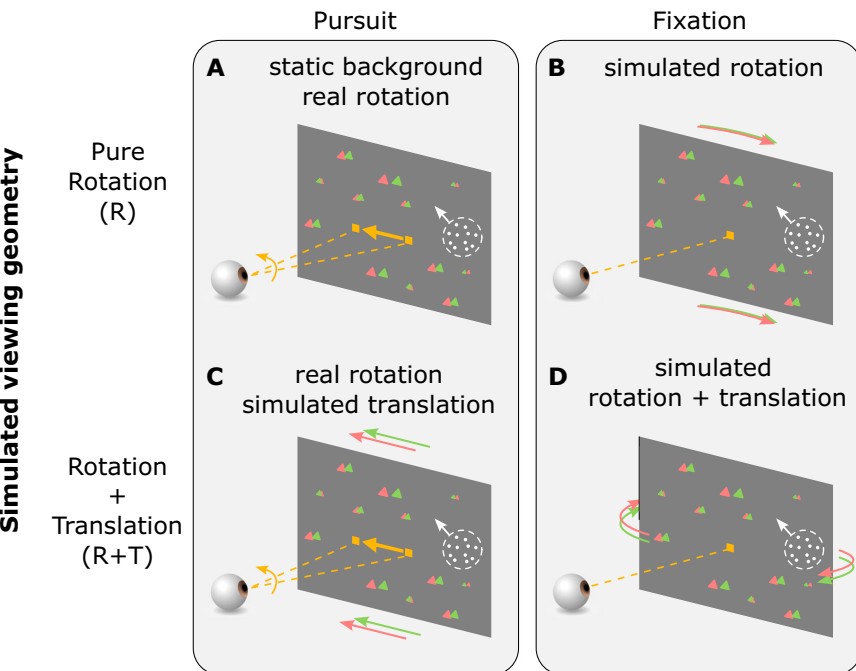

**Fig. 4 | Stimulus and task conditions for Experiment 1.** In the Pursuit conditions (**A**, **C**), the fixation target moved horizontally across the screen during the stimulus presentation period. Participants tracked the target by making smooth pursuit eye movements. In the Fixation conditions (**B**, **D**), the fixation target remained stationary at the center of the screen and participants maintained fixation on the target throughout the trial. In the R viewing geometry (**A**, **B**), a pure eye rotation was either executed by the participant in the Pursuit condition (**A**, yellow arrow) or simulated by background optic flow (**B**, red/green triangles). In the R+T viewing geometry (**C**, **D**), lateral translation of the eye relative to the scene was always simulated by background optic flow, and eye rotation was either real or simulated, as in the R geometry.

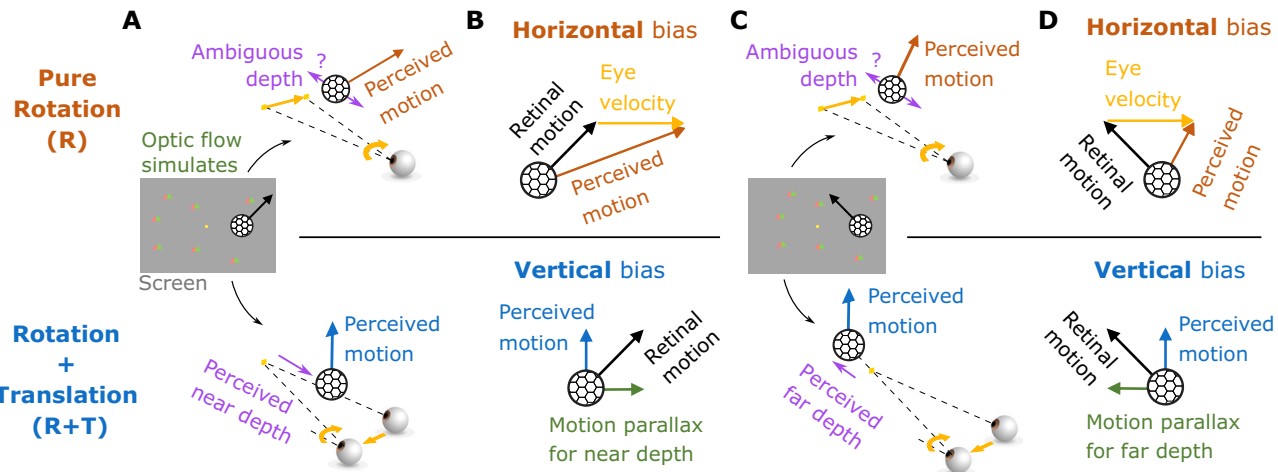

**Fig. 5 | Predictions for motion and depth perception in the R and R+T viewing geometries. A** Center, stimulus display showing a fixation target (yellow square) and an object (soccer ball shape) moving up and to the right (illustration of the Fixation condition). Meanwhile, background dots (red/green triangles) simulate the R (top) or R+T (bottom) viewing geometries. Top, in the R geometry, a rightward eye velocity (yellow arrow) is added to the image motion of the object, resulting in a rightward bias in motion perception (orange arrow). The object's depth remains ambiguous due to the absence of reliable depth cues. Bottom, in the R+T geometry, the horizontal component of the object's image motion can be explained away as motion parallax, such that the object is perceived as having a near depth. The residual vertical motion is then perceived as the object's motion in the world (blue arrow), leading to a vertical bias in perception. **B** Summary of the relationships between retinal motion, eye velocity, and perceived object motion in the R (top) and R+T (bottom) geometries. **C** Same format as **A** except that the object moves up and to the left on the display (center panel). In the R geometry (top), depth remains ambiguous, and the same rightward eye velocity is added to the object's motion, again producing a rightward bias. In the R+T geometry (bottom), the horizontal component of the object's image motion reverses direction, thus causing a far-depth percept. The perceived motion remains vertical. **D** Same format as **B**, except for the scenarios depicted in (**C**).

image motion on the screen when there was no background motion or pursuit eye movements (Fig. 6D and Supplementary Fig. S3, gray dots and squares). In the Pursuit condition with no background dots, participants showed a horizontal perceptual bias, suggesting a partial coordinate transform toward world coordinates (Fig. 6E, gray). This result is consistent with previous findings that motor commands associated with real pursuit modulate motion perception[12,15,22]. For most participants, the overall response pattern in the R+T Pursuit

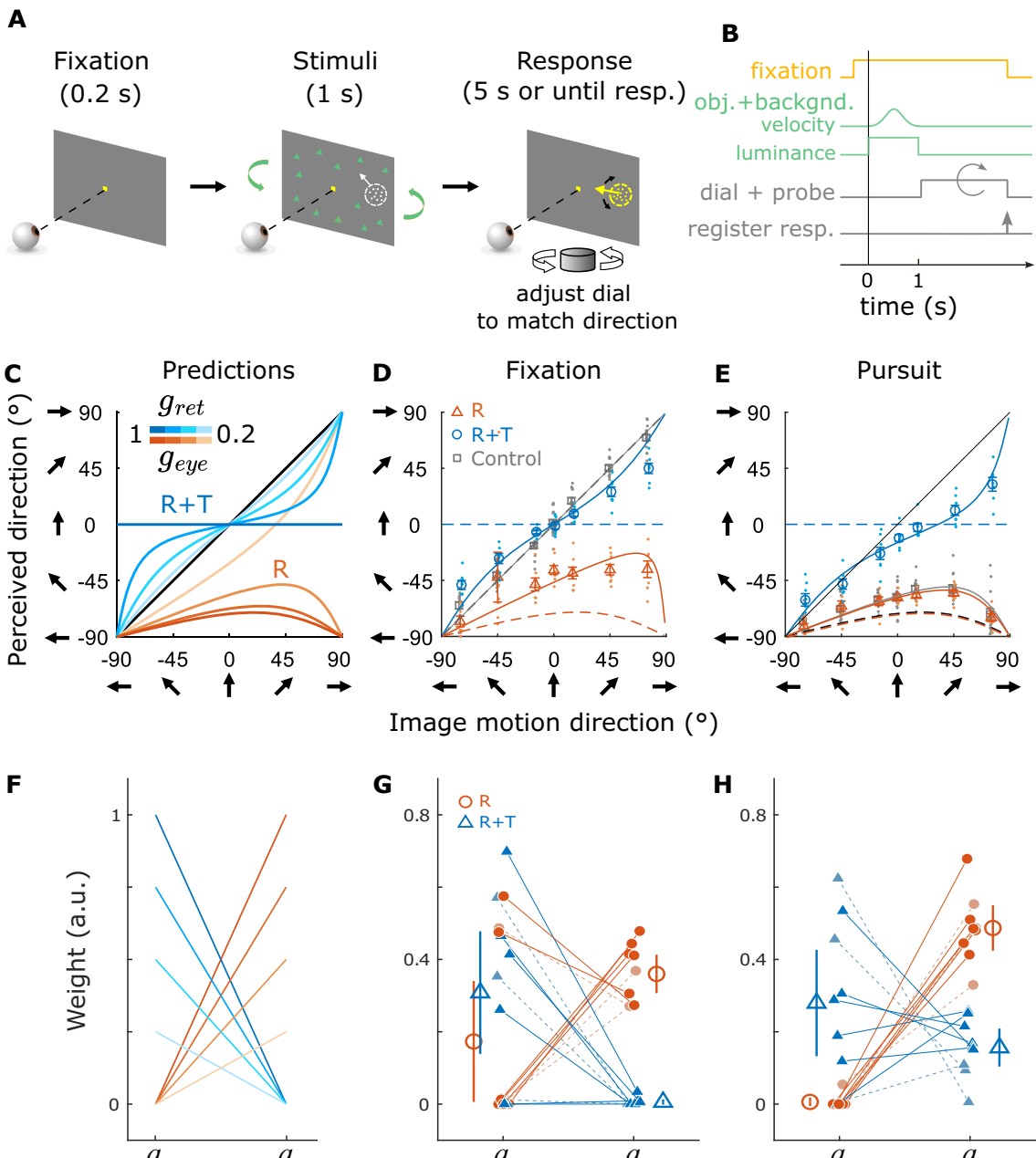

**Fig. 6 | Procedure, predictions, and results for the motion estimation task. A** At the beginning of each trial, a fixation target (yellow square) was presented at the center of the screen. After fixation, the visual stimulus appeared, including the background dots (green triangles) and the moving object (white dots). The 1-s stimulus presentation was followed by a response period, during which a probe stimulus composed of random dots (yellow dots) appeared, and the participant turned a dial to match the probe's motion direction with the perceived direction of the object. **B** Time course of stimulus and task events for the motion estimation task. **C** Prediction of perceived motion direction in the R (orange curves) and R+T (blue curves) geometries. In the R+T geometry, we expect a bias toward the vertical direction (0° on the *y*-axis), whereas a horizontal bias (toward -90° on the *y*-axis) is predicted in the R geometry. The color saturation of the curves depends on the gains in Eq. 9 ($g_{ret}$ and $g_{eye}$, respectively). Data from one example participant (h201)

in the Fixation (**D**) and the Pursuit (**E**) conditions. Individual dots represent the reported direction in each trial, and open markers indicate averages. Dashed curves are the predictions of the R (orange) and R+T (blue) geometries with $g_{ret} = g_{eye} = 1$, and solid curves are linear model fits to the data. Error bars indicate 1 SD ($N = 7$ trials). Note that data for downward image motion and rightward eye movements were folded and flipped, respectively, to generate the data representations in **C**–**E** (see Supplementary Fig. S1 for details). **F** Predictions of the weights for retinal and eye velocities in the R (orange) and R+T (blue) geometries. Shading indicates the gains as in (**C**). **G** Weights of retinal and eye velocities in the Fixation condition. Each filled circle and triangle represents data from one participant, and open symbols indicate means across participants. Dashed lines represent non-naïve participants. Error bars show 95% CIs ($N = 9$ participants). **H** Weights in the Pursuit condition ($N = 9$ participants). Format as in (**G**).

condition deviated substantially from the identity line and shifted toward the lower half of the plot (Fig. 6E and Supplementary Fig. S3C, D, I–L). We shall return to this observation below.

Because Eq. 2 shows that perceived object motion can be expressed as a linear combination of retinal and eye velocities in both

R and R+T geometries, we fit a linear model to each participant's responses in these experimental conditions. The linear model incorporated two parameters: a weight $a_{ret}$, which scales down the horizontal component of retinal velocity, and a weight $a_{eye}$, which accounts for the addition of eye velocity to retinal velocity. If $a_{ret} = 0$,

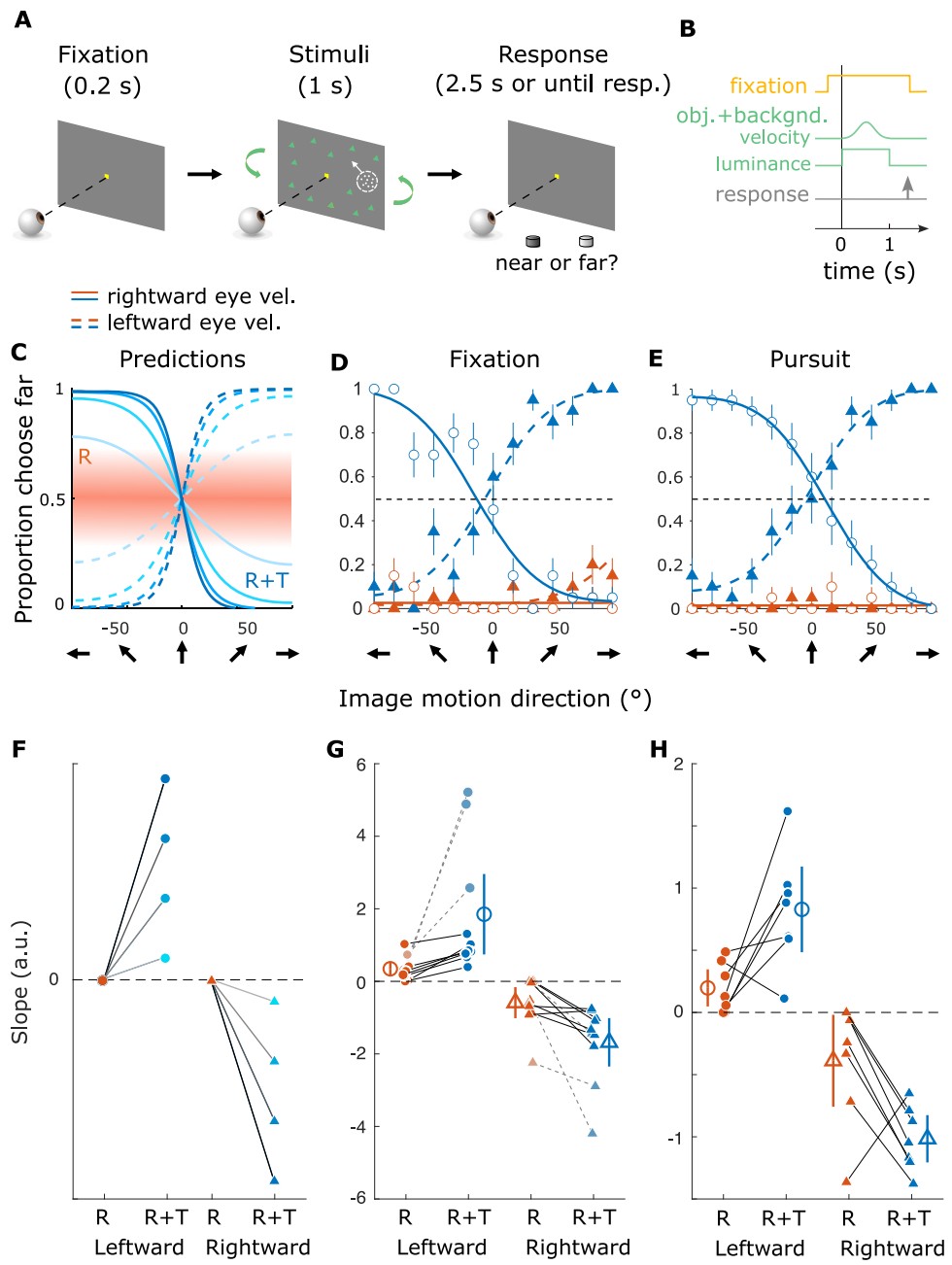

**Fig. 7 | Procedure, predictions, and results for the depth discrimination task.** **A** In each trial, a fixation point was followed by presentation of the visual stimuli, including the object and background. During or after stimulus presentation, participants pressed one of two buttons to report whether the object was perceived to be near or far relative to the fixation point. **B** Time course of stimulus and task events for the depth discrimination task. **C** Model predictions for depth perception in the R+T (blue curves) and R (orange) viewing geometries. Dashed and solid curves indicate leftward and rightward (real or simulated) eye rotations, respectively. Color saturation of the blue curves indicates the amount of eye movement accounted for in the prediction (same as Fig. 6C). Orange band represents the ambiguity of depth in the R viewing geometry. Psychometric curves from a naïve participant (h507) in the Fixation (**D**) and Pursuit (**E**) conditions. Error bars indicate S.E.M. (N = 20 trials) **F** Predicted slopes of psychometric functions in each simulated viewing geometry. Circles and triangles denote slopes for leftward and rightward eye movements, respectively. Orange and blue symbols represent the R and R+T viewing geometries, respectively. Saturation of symbol color indicates the gains in Eq. 10, same as in (**C**). **G** Slopes of psychometric functions in the Fixation conditions for all participants. Each solid symbol represents one participant, and large open markers show the group average. Error bars indicate 95% CIs (N = 10 participants). Dashed lines represent non-naïve participants. **H** Slopes of psychometric functions in the Pursuit conditions (N = 7 participants). Format as in (**G**).

the horizontal component of retinal velocity is attributed to object motion (Fig. 6F, orange). In contrast, if $a_{ret} > 0$, this suggests that part of the horizontal component is interpreted as MP (Fig. 6F, blue). The parameter $a_{eye}$ determines the extent to which eye velocity is added to the retinal velocity (Eq. 10; see "Methods" for details). Overall, this

simple linear model nicely captured the response patterns across individual participants (Fig. 6D, E and Supplementary Fig. S3; solid curves).

Equation 2 predicts that in the R geometry, $a_{ret}$ should be close to zero because retinal motion of the object can be fully explained as

object motion in the world, and $a_{eye}$ should be greater than zero because eye velocity is added to retinal velocity to achieve CT (Fig. 6F, orange lines). In the R+T geometry, because at least a portion of the horizontal retinal motion can be explained away as depth from MP, $a_{ret}$ should be greater than zero such that only part of the horizontal retinal velocity is perceived as object motion, whereas $a_{eye}$ should be close to zero because eye velocity would not be added to the object's motion (Fig. 6F, blue lines).

For most participants, the estimated weights align reasonably well with these predictions (Fig. 6G, H). We found greater values of $a_{ret}$ in the R+T condition than the R condition, and greater values of $a_{eye}$ in the R condition than the R+T case. This pattern is consistent across the majority of the participants and between the Pursuit and Fixation conditions ($p < 0.001$ for 8 out of 9 participants, 88.9%, in the Pursuit condition and for 5 out of 9 participants, 55.6%, in the Fixation conditions, two-sided Wilcoxon signed-rank test on 500 bootstrapped resamples with replacement for each participant; see Supplementary Table S1 for individual-level statistics). At the group level, $a_{ret}$ had a significantly greater value in R+T compared to the R condition during pursuit (Δmedian = 0.288, Hodges-Lehmann 95% CI = [0.00, 0.57], $Z = 2.55$, $p = 0.008$, effect size $r_W = 0.85$, two-sided Wilcoxon signed-rank test) but not during fixation (Δmedian = 0.261, HL 95% CI = [−0.48, 0.70], $Z = 0.77$, $p = 0.496$, $r_W = 0.26$, two-sided Wilcoxon signed-rank test); $a_{eye}$ had a significantly greater value in the R condition compared to R+T condition during both pursuit (Δmedian = 0.334, HL 95% CI = [0.19, 0.51], $Z = 2.67$, $p = 0.004$, $r_W = 0.89$, two-sided Wilcoxon signed-rank test) and fixation (Δmedian = 0.365, HL 95% CI = [0.27, 0.48], $Z = 2.67$, $p = 0.004$, $r_W = 0.89$, two-sided Wilcoxon signed-rank test). It is worth noting that there is considerable variability across individuals, most notably three participants (including one non-naïve participant) who are outliers in the R geometry during fixation (Fig. 6G, orange lines). This variability might be due to participants' varying ability to infer viewing geometry from optic flow, biases in estimating eye velocity, and/or variability in pursuit execution.

As noted above, in the R+T viewing geometry, many participants showed an overall bias towards the lower half of the plot in the Pursuit condition (Fig. 6E and Supplementary Fig. S3C, D, I–L, blue). Although this pattern deviates from our predictions (Fig. 6C), it is effectively captured by values of $a_{eye}$ (Fig. 6H, blue; Supplementary Fig. S2A, purple) that are greater than zero in our linear model fit for the Pursuit condition, which may suggest that some participants still perform a partial CT in the R+T geometry. This might indicate that these participants interpreted the viewing geometry as a mixture between R and R+T (see Discussion). Importantly, the presence of a nonzero $a_{eye}$ in the R+T geometry does not change our predictions for the depth perception task (Supplementary Fig. S2B), as described below.

## Viewing geometry biases depth perception based on motion parallax

The findings of the previous section demonstrate that motion perception of most participants is systematically biased by viewing geometry, in agreement with our theoretical predictions. Our analysis of viewing geometry, as described by Eq. 2, also makes specific predictions for how depth perception should vary between the R and R+T viewing geometries. In the case of R, the optic flow is depth invariant, the object was viewed monocularly, and the size of dots was kept constant across conditions. Therefore, there was no information available to form a coherent depth percept. When asking participants to judge whether the object was near or far compared to the fixation plane, we expect the response to be at chance or biased to an arbitrary depth sign based on their prior beliefs (Fig. 5A, C, top-right, purple; Fig. 7C, orange band). In the R+T geometry, we expect the horizontal component of the object's retinal motion to be explained away as MP (Fig. 5A and C, bottom-right, purple). Therefore, the ratio between the horizontal component of retinal motion and the simulated eye velocity

should determine the object's depth, based on the motion-pursuit law[35]. As retinal direction of the object changes, the horizontal component of its retinal motion varies from negative to positive, yielding a change in perceived depth from near to far, or vice versa. When asked to judge the depth sign (i.e., near or far), we expect participants to show inverted psychometric curves for opposite directions of eye movement, since perceived depth sign depends on both the sign of retinal velocity and the sign of eye velocity (Fig. 7C, blue curves).

In Experiment 2, we tested these predictions for perceived depth by asking human participants to discriminate the object's depth in the two viewing geometries (Fig. 7A, B). The stimuli were the same as in Experiment 1, except for the following: (1) the retinal direction of the object ranged from −90° to 90° instead of −90° to 270° and (2) after stimulus presentation, participants reported the perceived depth of the object (near or far relative to the fixation point) by pressing one of two buttons corresponding to each percept. Figure 7D, E show the results from one participant. In the R condition, the participant performed poorly, almost always reporting the object to be near, and no systematic difference was found between the two directions of eye movement (Fig. 7D, E, orange). Conversely, in the R+T condition, we observed a clear transition in depth reports from near to far, or vice-versa, as a function of retinal motion direction, and the psychometric curve inverted when the direction of eye movement was reversed (Fig. 7D, E, blue), consistent with our predictions (Fig. 7C).

Supplementary Fig. S4 shows similar data from other participants. Some participants show substantial non-zero slopes in the R condition, but almost always with lower magnitudes than in the R+T condition. Because the two viewing geometries were randomly interleaved within each session, we speculate that some participants might learn the inherent association between eye movement direction, retinal motion, and depth sign in the R+T viewing geometry and might generalize this association to the R viewing geometry (e.g., Supplementary Fig. S4G). Critically, because the retinal image motion of the object was identical between the R and R+T conditions, differences in depth perception can only be explained by the difference in optic flow patterns between the two viewing geometries.

The observed patterns of results are broadly consistent across most participants, as summarized in Fig. 7F–H. We observed a significantly greater magnitude of slope in the R+T geometry, as compared to the R geometry, for 8 out of 10 (80%) participants in the Fixation condition and 6 out of 7 (85.7%) participants in the Pursuit condition ($p < 0.05$, permutation test for each participant). At the group level, the magnitude of the slope of the psychometric function is greater in the R+T geometry compared to the R geometry, indicating a stronger depth percept (Δmedian |slope| = 0.874, HL 95% CI = [0.20, 4.12], $Z = 2.80$, $p = 0.002$, $r_W = 0.89$ for the Fixation condition; Δmedian |slope| = 0.750, HL 95% CI = [0.51, 1.23], $Z = 2.03$, $p = 0.046$, $r_W = 0.77$ for the Pursuit condition; two-sided Wilcoxon signed-rank test on the absolute slopes pooled across leftward and rightward eye movement directions). The distinct results between the R and R+T viewing geometries suggest that the main contribution of optic flow to depth perception observed here is generated by the combination of translation and rotation of the eye, which produces optic flow with a rotation pivot point at the fixation target. These results also provide direct evidence that humans automatically perceive depth from MP when eye rotation is inferred from optic flow, as proposed by ref. 52 (also see ref. 60).

Together, the results of Experiments 1 and 2 suggest that human participants automatically, and without any training, infer their viewing geometry from optic flow and subsequently perform the more natural computation in each geometry (CT for the R geometry, and depth from MP for the R+T geometry). As a result, the interaction between retinal and eye velocity signals automatically switches from summation (for CT) to division (for depth from MP) based on the inferred viewing geometry.

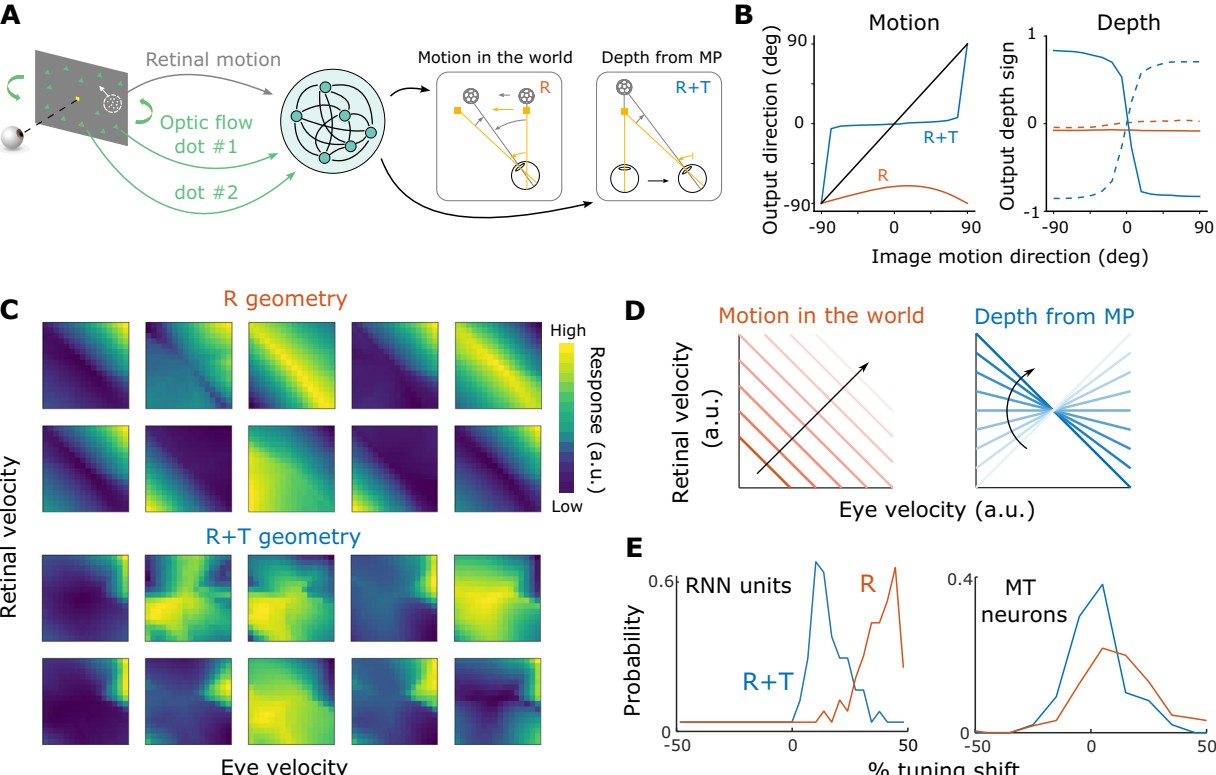

**Fig. 8 | Recurrent neural network trained to perform motion and depth computations. A** Inputs and outputs of the network. The network receives three inputs—retinal motion of the target object and the image motion of two background dots (one near and one far relative to the fixation point)—and it produces two outputs, object motion in the world and depth from MP. **B** Outputs of the trained RNN resemble human behavior. Left, The relationship between input retinal direction and the network's estimated motion direction in R+T (blue) and R (orange) geometries. Right, Relationships between estimated depth sign and retinal direction. Dashed and solid curves: leftward and rightward eye movement, respectively. **C** Joint velocity tuning for retinal and eye velocities in the R (top) and R

+T (bottom) geometries for 10 example recurrent units. Corresponding units are shown for both geometries. **D** Motion and depth computations require distinct joint representations of retinal and eye velocities. Left: velocity in world coordinates (orange lines) increases along the diagonal direction indicated by the black arrow. Right: depths from MP are represented by lines with varying slopes. **E** Histograms showing distributions of tuning shifts observed in RNN units (left) and MT neurons (right; adapted from ref. 57) for the R (orange) and R+T (blue) viewing geometries. A shift of 0% indicates a retinal-centered representation and a shift of 100% indicates a world-centered representation.

## Underlying neural basis implied by task-optimized recurrent neural network

So far, we have demonstrated that the visual system flexibly computes motion and depth based on optic flow cues to viewing geometry. How does the brain adaptively implement these computations? Recurrent neural networks (RNNs) have proven to be a useful tool for answering this type of question[61–64]. RNNs trained on specific tasks could perform these tasks with precision comparable to humans and showed response dynamics resembling those observed in biological systems[61–64].

We trained an RNN with 64 recurrent units to perform the motion estimation and depth discrimination tasks given retinal image motion of a target object and different optic flow patterns (Fig. 8A). At each time point, recurrent units in the network receive three inputs: retinal motion of the object and optic flow vectors of two background dots located at different depths. Because only horizontal rotation and translation of the eye were tested in our psychophysical experiments, the inputs to the network only included the horizontal velocity of the object's retinal motion and two optic flow vectors. Two scalar outputs are produced: the horizontal velocity of world-centered object motion and depth relative to the fixation point. We chose these inputs and outputs to approximate the structure of the psychophysical task while keeping the network architecture simple. Because of the layout of the rotational pivot in the R and R+T geometries (Fig. 2A, B), the optic flow vectors of two background dots, one at a near depth and the other at a far depth, suffice for distinguishing the two geometries. These two

dots move in the same direction in the R geometry, and move in opposite directions in R+T geometry (Fig. 2A, B). Notably, the viewing geometry (R vs. R+T) is not given directly to the network; it must infer the viewing geometry (and eye velocity) from optic flow and compute the output variables accordingly.

After training, the network's behavior in both motion and depth perception tasks replicates the basic patterns observed in human data, showing a horizontal directional bias in the R condition, a vertical motion bias in the R+T condition, a nearly flat psychometric function for depth discrimination in the R condition, and robust depth discrimination performance that depends on eye direction in the R+T condition (Fig. 8B, compare with Figs. 6C–E and 7C–E). In addition, we found that recurrent units in the RNN show different joint tuning profiles for retinal velocity and eye velocity depending on the simulated viewing geometry. Specifically, a negative diagonal structure is more pronounced among recurrent units in the R geometry than in the R+T geometry (Fig. 8C). This observation is consistent with the fact that the addition computation in the R context corresponds to a negative diagonal in the 2D joint velocity space (Fig. 8D, left), whereas the division computation in the R+T geometry corresponds to lines with different slopes (Fig. 8D, right). If neurons (or RNN units) are selective to a particular velocity in world coordinates, their joint velocity tuning would show a ridge along the negative diagonal (Fig. 8C, top and 8D, left); if neurons are selective to both a particular retinal motion and depth, their joint tuning would be a blob in the 2D space, as shown previously for some MT neurons[57,65] (Fig. 8C, bottom).

We quantified the extent of the diagonal structure in these RNN units by examining the asymmetry in the 2D Fourier transform of the data (see "Methods"), and we compared the results with those found in MT neurons by ref. 57. Specifically, the percentage of tuning shift in the RNN units was measured as the normalized product of inertia in the 2D Fourier space (see "Methods" for details). We found that RNN units exhibit substantially more diagonal structure in the R geometry than R+T ($\Delta$median = 24.69, $U$ = 119, $Z$ = 9.19, $p$ < 0.001, $r_W$ = 0.81, HL 95% CI = [0.79, 40.12], two-sided Mann–Whitney U test; Fig. 8E, left). Xu & DeAngelis[57] modeled MT neurons' responses to motion parallax stimuli under different conditions. While the experimental conditions were not the same as the R and R+T geometries discussed in this study, they have some similar features. When only retinal motion and eye movement were present, and no background optic flow was shown, the viewing geometry is likely to be more consistent with the R viewing geometry, in which a horizontal eye velocity is added to the retinal image motion. This is supported by similar patterns of results between the control and the R viewing geometry in the Pursuit condition of Experiment 1 (Fig. 6E, gray dots versus orange dots). Xu & DeAngelis[57] observed a significant shift in the tuning of MT neurons toward world coordinates during pursuit (median = 12.20, $Z$ = 7.83, $p$ < 0.001, one-tailed Wilcoxon signed-rank test; Fig. 8E, right, orange). In contrast, when the animal fixated at the center of the screen, and optic flow simulated the R+T geometry, the extent of the diagonal shift was significantly reduced ($\Delta$median = −11.40, $Z$ = −5.360, $p$ < 0.001, one-tailed Wilcoxon rank sum test; Fig. 8E, right, blue). These observations broadly align with our findings in RNN units, suggesting a potential role of MT neurons in flexibly computing object motion and depth under different viewing geometries.

## Discussion

We demonstrate that the traditional model of visual compensation for pursuit eye movements, based on vector subtraction, fails to generalize to even the simplest combinations of eye translation and rotation. Instead, we provide a theoretical framework that relates object motion and depth (in the world) to retinal and eye velocities of a moving observer, across a range of possible viewing geometries. This framework unifies two well-known perceptual phenomena—coordinate transformation and computation of depth from motion parallax—that have generally been studied separately. We generated theoretical predictions for how perception of object motion and depth should depend on viewing geometry simulated by optic flow, and we verified these predictions using a series of well-controlled psychophysical experiments. Our results suggest that humans automatically, without any feedback or training, infer their viewing geometry from visual information (i.e., optic flow) and use this information in a context-specific fashion to compute the motion and depth of objects in the world. They flexibly attribute specific components of image motion to either eye rotation or depth structure, depending on the inferred viewing geometry. A recurrent neural network trained to perform the same tasks shows underlying representations that are somewhat similar to neurons in area MT, suggesting a potential neural implementation of the flexible computations of motion and depth.

In the traditional view of visual perception during eye movements, sensory consequences of self-generated actions are considered detrimental to perception and, therefore, should be suppressed[66]. By contrast, our study demonstrates that humans utilize the visual consequences of smooth pursuit eye movements (i.e., optic flow) to infer their viewing geometry and adaptively compute the depth and motion of objects. It is precisely the sensory consequences of self-motion that provide rich information about the relationship between the observer and the dynamic 3D environment, allowing more accurate perception of the 3D world during active exploration[67–69]. Importantly, our framework does not contradict the traditional model of visual compensation for pursuit eye movements. Rather, the conventional compensatory mechanism is subsumed within our

framework under a specific context (i.e., the R geometry). For example, our linear model (Eq. 10) explains the Filehne illusion and Aubert–Fleischl phenomenon in the R geometry with $a_{eye}$ < 1, which indicates an underestimate of eye velocity. Additionally, the CT computation within our framework provides an explanation for the observed reduction in motion detection sensitivity when an object moves in the direction opposite to smooth pursuit in the R geometry[70].

In our framework, the information about viewing geometry provided by optic flow depends only on how the eye moves relative to the scene. This relative motion may arise from movement of the entire body, the head, rotation of the eye in its socket, or combinations of these actions. It is possible that humans and other species deliberately organize eye and head movements to produce optic flow informative for discerning the 3D layout of the environment. For example, studies in mice[71], pigeons[72], and locusts[73] have demonstrated that these animals actively move their head to generate motion parallax cues for depth perception. It has also been shown that involuntary head movements at the microscopic level provide sufficient information for humans to compute depth from MP[74]. However, how the movements of eye, head, and body are coordinated and optimized for 3D active sensing is beyond the scope of our current framework, and remains an interesting question for future research.

### Interactions between object motion and depth

In our experiments, we presented the object only to one eye and kept its size constant, such that the object's depth was ambiguous and subject to influence from motion information. Presumably, each participant's prior distribution over possible object depths might affect their depth perception differently[75–77]. This might account for some of the cross-participant variability observed in the depth discrimination task. To directly test the interactions between motion and depth, a future direction would be to include additional depth cues for the object and to examine how this affects perceived motion direction. When different levels of depth cues (e.g., congruent vs. incongruent with the MP cue) are introduced, we might expect different amounts of retinal motion to be explained away as depth from MP. For example, if binocular disparity cues always indicated that the object was in the plane of the visual display, we would expect a smaller vertical bias in perceived direction in the R+T geometry, as one should not attribute a horizontal component of image motion to depth.

It is also worth noting that in the R+T geometry, a greater value of $a_{ret}$ should lead to a greater perceived depth magnitude (Eq. 8). However, we did not find a significant correlation between $a_{ret}$ measured in Experiment 1 and the slope of the psychometric functions measured in Experiment 2 ($r$ = −0.182, $p$ = 0.639 for the Fixation condition; $r$ = −0.330, $p$ = 0.524 for the Pursuit condition, Pearson's correlation). This might be because Experiment 2 was designed to measure the discriminability of depth signs rather than magnitudes. Participants are likely to exhibit individual variability in depth-sign discrimination thresholds that is independent of their perceived depth magnitude, such as variations in internal noise on representations of retinal or eye velocity. Future work that simultaneously measures perceived motion direction and depth magnitude would provide valuable insights into the relationship between motion and depth magnitude estimation on a trial-by-trial basis.

In addition, Eq. 2 shows that object motion and depth are underdetermined in the absence of other depth cues, even when the viewing geometry is unambiguous. In the R+T geometry, the motion-pursuit law applies only when the object is stationary (Eq. 4). When $p'$ = 1 and $\omega_{obj} \neq 0$, object motion must be subtracted from retinal image motion in order to accurately compute depth from MP:

$$d' = \frac{\omega_{ret} - \omega_{obj}}{\omega_{eye}}. \tag{5}$$

If the observer mistakenly believes the object is stationary and does not perform this subtraction, a bias in depth perception occurs. Indeed, this effect of object motion on depth perception was demonstrated in a recent study[78].

A related line of research has investigated the phenomenon of optic flow parsing, namely the process of inferring object motion from optic flow[40,46,58,79–82]. Warren and Rushton[79] found that humans can differentiate between optic flow patterns generated by eye rotation (i.e., the R geometry) and lateral translation, showing that depth modulates object motion perception only in the latter case, as expected from the 3D geometry. Our study demonstrates that humans extract information about depth structure only when optic flow is depth-dependent, revealing a flexible and automatic switch between attributing the source of retinal motion to object motion vs. depth.

Previous studies have demonstrated the role of dynamic perspective cues (which are associated with rotation of the scene around the fixation point in the R+T geometry) in stabilizing the scene and disambiguating depth sign[32,60,83], and neural correlates of depth coding from these visual cues have been found in macaque area MT[52]. Our Experiment 2 provides direct evidence that humans automatically perceive depth from dynamic perspective cues in the R+T viewing geometry, with or without corresponding pursuit eye movements. Our findings thus add important insights into the visual processing of depth from MP.

## Compensation for pursuit eye movement depends on viewing geometry

The perceptual consequences of pursuit eye movement have been extensively studied in the past decades[5,9–12,15,16,19–21,23–25,28,84–86]. These studies often hypothesize that the brain generates a reference signal related to eye velocity and subtracts it from the retinal image motion in order to perceive a stable world[12,15] (Fig. 1C). While this line of research has successfully explained many visual phenomena that involve pure eye rotations in 2D displays, it does not generalize to situations in which there is 3D scene structure and combinations of eye translation and rotation (Fig. 2). Eye translation introduces components of optic flow that are depth-dependent, such that one cannot perform a simple vector subtraction to compensate for the visual consequences of smooth pursuit. Our study provides a much more general description of how the brain should compute scene-relative object motion and depth under a variety of viewing geometries that involve combinations of eye translation and rotation.

Importantly, we observed similar patterns of perceptual biases in the Fixation and Pursuit conditions, suggesting that the different effects of viewing geometry on motion and depth perception cannot simply be explained by retinal slip caused by imperfect pursuit eye movements. Interestingly, in the Pursuit conditions, the effects are similar between the R condition and the no background condition (Supplementary Fig. S3, orange vs. gray markers), indicating that either real pursuit alone or optic flow alone can shift perceived object direction toward the eye movement. This suggests an absence of additive effects between extra-retinal signals and optic flow in the CT computation. However, our findings should not be taken to imply that optic flow always serves as a useful proxy for non-visual sources of information about eye movements. Extra-retinal signals related to eye movement, such as efference copy and proprioception, play crucial roles in visual processing that may not be replaced by optic flow[87–89]. Our experiments, however, were not designed specifically to quantify the relative contributions of optic flow and extra-retinal signals to inferring viewing geometry, and this would be a valuable topic for further research.

A few previous studies have investigated the interactions between motion and depth perception during self-motion. For example, Wallach et al.[90] showed that underestimating an object's depth resulted in an illusory rotation of the object during lateral head translation. Gogel

and colleagues investigated how this apparent motion changed as a function of under- or over-estimation of depth, showing that the direction of the perceived motion changed systematically based on the geometry of motion parallax[91,92]. In addition, illusory motion induced by head motion can be added to or subtracted from the physical movement of objects[83,93,94].

In these studies, observers were explicitly instructed to laterally translate their heads[95]. As a result, the viewing geometry was always unambiguous, and the role of different geometries was not explored. The source of uncertainty, or errors, was thought to be either the intrinsic underestimation of distance in a dark room[90,96,97] or that induced by binocular disparity cues[92]. How does the observer's belief about the viewing geometry modulate the interaction between motion and depth? What are the cues (e.g., optic flow and extraretinal signals) for disambiguating viewing geometry? These important questions have not been addressed by previous studies.

We demonstrate that humans use optic flow information to infer their viewing geometry and flexibly compute object motion and depth based on their interpretations of the geometry. Moreover, our work provides important insights into how the brain solves the causal inference problem of parsing retinal image motion into different causes—object motion in the world and depth from motion parallax—based on the information about self-motion given by optic flow.

## Recurrent neural network and the neural basis of contextual computation

Our RNN model provides insights into the neural basis of computing object motion and depth in different viewing geometries. By comparing representations in the network model with neurons in MT, we suggest a potential role of MT neurons in implementing flexible computations of motion and depth. Area MT has been linked to the perception of object motion[98–100], and perception of depth based on both binocular disparity[101,102] and MP cues[52,54]. Emerging evidence shows that sensory areas receive top-down modulations from higher cortical regions that reflect perceptual decision variables or cognitive states[103,104]. Therefore, we speculate that neurons in MT might receive feedback signals about viewing geometry from higher-level areas, such as the medial superior temporal (MST) area or areas in the intraparietal sulcus, and use these signals to modulate the response to retinal motion. A recent study has shown that neurons in dorsal MST are selective for large-field optic flow that simulates eye translation and rotation in the R+T geometry[105], which suggests a potential source of information about viewing geometry that is known to feed back to area MT[106–108]. Furthermore, a recent study[59] has demonstrated that responses of MT neurons are modulated by background optic flow in a manner that is consistent with perceptual biases that are associated with optic flow parsing (i.e., flow parsing[40,46,58,79–82]). However, whether activity in area MT will reflect flexible computations of object motion and depth, based on inferred viewing geometry, remains to be examined.

Another possible source of signals related to viewing geometry, as suggested by our RNN model, is the recurrent connections within area MT. Different optic flow patterns used in our study might differentially trigger responses of a subset of MT neurons whose receptive fields overlapped with the background dots, and these responses could, in turn, modulate MT neurons with receptive fields overlapping the object. As shown by ref. 57, a partial shift in the tuning preference observed in MT neurons, in theory, suffices for computing world-centered motion and depth. Further investigation with inactivation techniques would be desirable to determine whether or not higher-level cortical areas are involved in these flexible computations of motion and depth.

While the comparison between our RNN and the data from ref. 57 is suggestive, the conclusions that can be drawn remain speculative. New neurophysiological studies are needed, in which MT population

activity is measured in response to the same stimulus conditions used in this study, in order to more directly assess the mechanisms by which MT neurons might support the flexible computation of motion and depth based on viewing geometry.

### Limitations and future directions

Deriving from 3D geometric principles, our theoretical framework makes quantitative predictions about motion and depth perception during self-motion for a range of scenarios as depicted in Fig. 3. Perceptual biases, as predicted by our framework, were demonstrated for perception of object motion and depth in two simple viewing geometries: Pure Rotation (R) and Rotation + Translation (R+T). In addition, our framework also makes predictions for scenarios in which the viewing geometry is intermediate between R and R+T. Examination of these intermediate viewing geometries in future studies will provide further validation of our theory.

A limitation of our theory is that Eq. 2 only applies when both the observer and the pursuit target translate in the fronto-parallel plane, such that the relative position of the rotation pivot remains constant (Fig. 3). Because natural behavior involves movements along multiple axes at the same time[109], an extension of our theory is needed to better understand visual perception in natural environments. Further analysis of how optic flow is constrained by different viewing geometries might yield insights into a more generalized theory[1,26,110,111].

Although our psychophysical data broadly align with our theoretical predictions, they are not fully accounted for by Eq. 2. Specifically, the measured perceptual biases are typically partial biases. Here, we capture substantial deviations from the ideal predictions using discount factors analogous to a flow parsing gain[40]. However, multiple sources might contribute to these deviations: observation noise, uncertainty about the viewing geometry, underestimation of smooth pursuit eye velocity[10], cue conflict between vestibular and visual signals[112], a slow-speed prior[113], and so on. To fully quantify and understand the inference performed by the observer, a more comprehensive probabilistic model of motion and depth perception will be needed[114,115]. Specifically, the problem of differentiating between object motion, depth, and self-motion might be formalized as an instance of the Bayesian causal inference problem[2–4,116]. In addition, research has shown that humans use smooth pursuit to track perceived motion, such as motion aftereffects[117] and discrete spatial jumps[118], in the absence of actual retinal image motion. Our current framework, however, only considers the interactions between retinal motion and eye movements. How these second-order effects might be integrated into our theory remains an open question.

In our neural network simulations, the RNN was directly trained to reproduce the predicted motion and depth perception. Whether or not such perceptual biases naturally emerge in networks trained to estimate self-motion or encode videos of natural scenes is an interesting future direction to explore[119,120]. Our comparison of MT neural responses between the R and R+T conditions was indirect, as previous experimental work did not explicitly simulate these two viewing geometries[39,52,54,57]. An ongoing study that directly measures the responses of MT neurons in the two viewing geometries will provide important insights into the neural mechanisms underlying flexible computations of motion and depth.

## Methods

### Participants

Ten participants (4 males and 6 females, 18–58 years old) with normal or corrected-to-normal vision were recruited for the psychophysical experiments. All participants had normal stereo vision (<50 arcseconds, Randot Stereotest). Seven of the participants (h507, h508, h510, h512, h518, h520, and h521) were naïve to the experiments and unaware of the purpose of this study. All participants completed the depth discrimination task (Experiment 2) first, and nine of them finished the

motion estimation task (Experiment 1) in subsequent sessions. Participants provided informed written consent prior to data collection and were financially compensated for their time. The study was approved by the Institutional Review Board at the University of Rochester. No statistical method was used to predetermine the sample size. No data were excluded from the analyses.

### Apparatus

Participants sat in front of a 48-inch computer monitor (AORUS FO48U; width, 105.2 cm; height, 59.2 cm) at a viewing distance of 57 cm, yielding a field of view of ~85° × 55°. A chin and forehead rest was used to restrict participants' head movements. Position of each participant's dominant eye was monitored by an infrared eye tracker (Eyelink 1000Plus, SR-Research) positioned on a desk in front of the participant at a distance of ~52 cm. The refresh rate of the monitor was 60 Hz, the pixel resolution was 1920 × 1080, and the pixel size was ~ 2.6′ × 3′ arcmin. During the experiments, participants viewed the visual display through a pair of red-blue 3D glasses in a dark room. The mean luminance of the blank screen was 0 cd m$^{-2}$ (due to the OLED display), the mean luminance of the object (including dots and dark background within the aperture) was 1.383 cd m$^{-2}$, and the mean luminance of the background optic flow (including dots and dark background) was 0.510 cd m$^{-2}$. Commercial software (TEMPO, Reflective Computing, Olympia, WA) was used to control the trial sequence and record data.

### Stimuli

Visual stimuli were generated by custom software and rendered in 3D virtual environments using the OpenGL library in C++. Participants were instructed to fixate on a square at the center of the screen, and a random-dot patch (referred to as the "object"; diameter 5°) was presented on the horizontal meridian, interleaved between left and right hemifields, at 10° eccentricity. Viewing of the object was monocular to remove binocular depth cues. The size of the dots comprising the object was constant on the screen across conditions, to avoid providing depth cues from varying image sizes. In most conditions, a full-field 3D cloud of background dots was presented for the same duration as the object. The motion of the background dots was generated by moving the OpenGL camera, simulating either the R or R+T viewing geometries (Fig. 2A, B; see Supplementary Information for details). The movements of the object, background optic flow, and fixation target followed a Gaussian velocity profile (±3σ) spanning a duration of 1 s. The immediate region surrounding the object (2× the object's radius) was masked to avoid local motion interactions between background dots and the object.

**Simulating viewing geometry with optic flow.** A 3D cloud of background random dots simulated 4 different configurations of eye translation and/or rotation. (1) In the R Fixation condition (Supplementary Movie 3), the OpenGL camera rotated about the y-axis (yaw rotation) to track the moving fixation target such that it remained at the center of the screen; this resulted in rotational optic flow that simulated the R geometry, while requiring no actual pursuit eye movement. (2) In the R+T Fixation condition (Supplementary Movie 4), the OpenGL camera translated laterally while counter-rotating to keep the world-fixed fixation target at the center of the screen. This generated background optic flow that simulated both translation and rotation in the R+T geometry, while again requiring no smooth pursuit. (3) In the R Pursuit condition (Supplementary Movie 5), the OpenGL camera remained stationary. As a result, the background dots did not move on the screen. A fixation target appeared at the center of the screen and moved 3 cm, either leftward or rightward. The movement of the fixation target followed a Gaussian speed profile (±3σ) spanning 1 second, resulting in a peak speed of ~13° s$^{-1}$ and a mean speed of ~5.3° s$^{-1}$. Participants were required to track the fixation target with

their eyes. (4) In the R+T Pursuit condition (Supplementary Movie 6), the OpenGL camera translated laterally (leftward or rightward) by 3 cm in the virtual environment (following the same Gaussian speed profile). Therefore, the background dots appeared to translate in the opposite direction on the screen, providing optic flow that simulated eye translation. Throughout the trial, a fixation target appeared at a fixed location in the virtual environment but moved on the screen due to the camera's translation. The participant was required to make smooth eye movements to remain fixated on the world-fixed target. Note that background elements were triangles of a fixed size in the scene, such that their image size was inversely proportional to their distance (unlike for the object stimulus).

**Object motion.** A random-dot patch (the "object") with a fixed dot density of ~4.7 dots per degree and a dot size of ~0.2° was rendered monocularly to the right eye. To make the object's depth ambiguous, the dot size and diameter of the aperture were kept constant across all stimulus conditions. In each trial, the object moved as a whole (the aperture and the random dots within it moved together), in one of several directions on the screen. The position and motion trajectory of the object were carefully computed such that it yielded identical image motion between the R and R+T conditions (see Supplementary Information for details). The speed of the object followed a Gaussian profile with a maximum speed of ~6.67° s$^{-1}$ and a mean speed of ~2.67° s$^{-1}$.

**Experiment 1: Procedures and experimental conditions**
In Experiment 1, participants performed a motion estimation task. At the beginning of each trial, a fixation point appeared at the center of the screen, followed by the onset of the object and background dots. In each trial, the direction of retinal motion of the object was randomly chosen from −90° to 270° with 30° spacing (we defined the upward direction as 0°, and the angle increases in a clockwise direction). The object and background were presented for 1 s, after which another patch of dots (the "probe"; rendered binocularly at the same depth as the screen) appeared at the same location on an otherwise blank screen. Participants used a dial to adjust the motion direction of the probe such that it matched the perceived direction of the object. After adjusting the dial, participants pressed a button to register their response and proceeded to the next trial after an inter-trial interval of 1.5 s. Failure to register the response within 5 s from probe onset resulted in a time-out, and the trial was repeated at a later time. Eye position was monitored throughout each trial, and failure to maintain fixation within a ±5° rectangular window around the fixation target resulted in a failed trial, after which visual stimuli would be immediately turned off. Audio feedback was provided at the end of each trial to indicate successful completion of the trial with a high-pitched tone and failed trials (fixation break or time out) with a low-pitched tone. For completed trials, information about response error was not provided to the participants in any form. This lack of feedback prevented participants from learning to compensate for perceptual biases induced by optic flow.

Four main stimulus conditions were randomly intermixed within each session: two eye-movement conditions × two background conditions (Fig. 4 and Supplementary Movies 3–6). The two eye-movement conditions were: (1) the Pursuit condition, in which the participant visually tracked a fixation target that moved across the center of the screen while simultaneously viewing an object composed of random dots at 10° eccentricity; (2) the Fixation condition, in which participants fixated on a stationary target at the center of the screen while background dots simulated eye translation and/or rotation. The direction of actual or simulated eye movements was either 90° (rightward) or −90° (leftward), randomly interleaved across trials. The two background conditions were: (1) the R viewing geometry, in which the motion of the background dots was consistent with a pure eye rotation; (2) the R+T viewing geometry, in which the background dots

simulated a combination of lateral translation and rotation of the eye. In addition, two control conditions were interleaved with the main conditions, including Pursuit and Fixation conditions with object motion in the absence of background dots (Supplementary Movies 7 and 8).

Before the main experimental session, a practice session (72–144 trials) was completed to ensure a correct understanding of the task and to give participants practice with the dial-turning behavior. In this short block of practice trials, only the object was present, and no background was shown. All participants successfully reported the object's motion direction within a ±15° range around the ground truth before proceeding to the main experimental session.

**Experiment 2: Procedures and experimental conditions**
In Experiment 2, participants performed a depth discrimination task. The visual stimuli and experimental procedure were the same as in Experiment 1, except that participants pressed one of two buttons, either during the stimulus period or within 2.5 s afterward, to report whether the object was located near or far compared to the fixation point. Background and eye movement conditions were the same as those in Experiment 1. In each trial, the direction of retinal motion of the object was randomly chosen from −90° to 90° with 15° spacing. Because depth is expected to be determined by the horizontal component of retinal motion and eye velocity, we did not include directions in the range between 90° and 270°, which differ from −90° to 90° only in vertical components.

Before the formal experimental session, participants underwent a practice session to become familiar with the stimuli and the task. In the practice session, the background motion was the same as the R+T condition, and the object moved in horizontal directions at different speeds such that its retinal motion could be fully explained as depth from motion parallax. After 72 practice trials, the experimenter decided to either (1) proceed to the formal experimental session if the accuracy was above 95%, or (2) continue with another practice session in which the object was viewed binocularly to aid depth perception. After the binocular session, another monocular practice session was run to ensure that participants performed the task well above chance. Three participants did not proceed to the formal experimental sessions due to failure to report depth with an accuracy above 80% during the practice sessions.

**Data analysis**
**Analysis of eye-tracking data.** For most participants, eye position signals measured by the eye-tracker were used to ensure fixation behavior and to compute smooth pursuit gains. In the fixation conditions, trials with eye positions outside of a 10°-by-10° rectangular window around the fixation target for over 100 ms were excluded from the analysis. In the pursuit conditions, pursuit velocities were obtained by filtering the eye position data with a first-derivative-of-Gaussian window (SD = 25 ms), followed by a velocity threshold at 40° s$^{-1}$ and an acceleration threshold at 300° s$^{-2}$ to remove catch-up saccades and artifacts. The median of the pursuit velocity was obtained across trials, and the ratio between its peak and the peak velocity of the pursuit target was computed as pursuit gain. Across all participants, the mean pursuit gains are 0.755 (SE = 0.119) in Experiment 1 and 0.768 (SE = 0.0512) in Experiment 2 (Table 1). There was no significant difference between the pursuit gains in Experiments 1 and 2 (p = 0.535, Wilcoxon rank sum test). Pursuit gains were significantly lower in the R geometry compared to the R+T geometry (Δmedian = −0.12, HL 95% CI = [−0.30, 0.07], Z = −2.20, p = 0.031, $r_W$ = −0.83 for Experiment 1; Δmedian = −0.24, HL 95% CI = [−0.27, 0.06], Z = −2.03, p = 0.047, $r_W$ = −0.77 for Experiment 2; two-sided Wilcoxon signed-rank test). This finding is consistent with contextual effects on smooth pursuit eye movements reported previously[22]. Across participants, pursuit gain was not correlated with $a_{ret}$ and $a_{eye}$ in Experiment 1 (Pearson's r = 0.586, p = 0.097

**Table 1 | Pursuit gains of each participant in the R and R+T viewing geometries for Experiments 1 and 2**

| Participants | Experiment 1 | | Experiment 2 | |
|---|---|---|---|---|
| | R geometry | R+T geometry | R geometry | R+T geometry |
| h201 | / | / | / | / |
| h500 | / | / | / | / |
| h501 | 1.460 | 1.394 | / | / |
| h507 | 0.397 | 0.513 | 0.533 | 0.801 |
| h508 | / | / | 0.980 | 0.916 |
| h510 | 0.658 | 0.733 | 0.537 | 0.774 |
| h512 | 0.713 | 0.837 | 0.781 | 0.809 |
| h518 | 0.672 | 0.974 | 0.696 | 0.957 |
| h520 | 0.835 | 0.949 | 0.844 | 1.106 |
| h521 | 0.426 | 0.621 | 0.734 | 0.889 |
| Median | 0.672 | 0.837 | 0.734 | 0.889 |

Eye tracking was not conducted for h201 and h500 in Experiment 1, and the Pursuit conditions were not included in Experiment 2 for h201, h500, and h501. Participant h508 did not participate in Experiment 1.

for $a_{ret}$ and $r = -0.291$, $p = 0.447$ for $a_{eye}$ in the R geometry; $r = -0.392$, $p = 0.296$ for $a_{ret}$ and $r = 0.104$, $p = 0.789$ for $a_{eye}$ in the R+T geometry) or the magnitudes of slopes in Experiment 2 (Pearson's $r = -0.092$, $p = 0.844$ for the R geometry and $r = -0.528$, $p = 0.230$ for the R+T geometry).

**Behavioral data analysis.** In Experiment 1, because we expect the pattern of biases to be symmetric around the horizontal axis (Fig. 2 and Supplementary Fig. S1A, B), image motion directions and reported directions from trials in which the object moved in directions from 90° to 270° were folded around the horizontal axis and pooled with those from trials with retinal directions in the range of −90° to 90° (Supplementary Fig. S1A, C). Following a similar logic, we pooled data from trials with rightward and leftward eye movements by flipping the velocities about the vertical axis for rightward pursuit trials (Supplementary Fig. S1A–D). This results in a consistent leftward bias prediction in the R geometry shown in Fig. 6C. Data pooling was performed only for visualization, and model fitting utilized the full range of motion.

Because of imperfect pursuit eye movements by the participants, the actual retinal image motion of the object was contaminated by retinal slip. It differed from the intended velocity in the Pursuit conditions. We corrected this by factoring in the measured pursuit gain, $g_{pursuit}$, for each participant:

$$\widetilde{\omega}_{eye}^{x} = g_{pursuit}\omega_{eye}^{x}, \tag{6}$$

$$\widetilde{\omega}_{ret}^{x} = \omega_{ret}^{x} + \omega_{eye}^{x} - \widetilde{\omega}_{eye}^{x} = \omega_{ret}^{x} + (1 - g_{pursuit})\omega_{eye}^{x}, \tag{7}$$

where $\widetilde{\omega}_{eye}^{x}$ and $\widetilde{\omega}_{ret}^{x}$ are the horizontal components of the real eye and retinal velocities, respectively; $\omega_{eye}^{x}$ and $\omega_{ret}^{x}$ are the intended horizontal components of eye and retinal velocities, respectively. Because pursuit eye movements were always along the horizontal axis, the vertical components of the velocities were unaffected. Eye velocities are defined relative to a fixed point in the world, and retinal velocities are relative to the eye.

In the R+T viewing geometry, we assumed that a portion of the horizontal component of retinal image velocity would be explained as motion parallax for computing depth:

$$\widehat{d'} = \frac{g_{ret}\widetilde{\omega}_{ret}^{x}}{g_{eye}\widetilde{\omega}_{eye}^{x}}, \tag{8}$$

where $\widehat{d'}$ is the perceived relative depth and $g_{ret}$ represents the proportion of horizontal retinal motion perceived as motion parallax. Similarly, we assumed that a portion of eye velocity was accounted for by a factor, $g_{eye}$, in the R viewing geometry. Therefore, Eq. 2 can be rewritten as:

$$\omega_{obj}^{x} = \widetilde{\omega}_{ret}^{x} + (1 - (1 + \frac{g_{ret}\widetilde{\omega}_{ret}^{x}}{g_{eye}\widetilde{\omega}_{eye}^{x}})p')g_{eye}\widetilde{\omega}_{eye}^{x}$$
$$= (1 - g_{ret}p')\widetilde{\omega}_{ret}^{x} + (1 - p')g_{eye}\widetilde{\omega}_{eye}^{x}. \tag{9}$$

This formula indicates that perceived object motion is a linear combination of retinal and eye velocities, with varying weights on each velocity term that depend on the viewing geometry, $p'$. Simplifying this equation, we used a linear model to capture this relationship:

$$\omega_{obj}^{x} = (1 - a_{ret})\widetilde{\omega}_{ret}^{x} + a_{eye}\widetilde{\omega}_{eye}^{x}, \tag{10}$$

$$\text{where } a_{ret} \triangleq g_{ret}p', \tag{11}$$

$$a_{eye} \triangleq (1 - p')g_{eye}. \tag{12}$$

In the R geometry, $p' = 0$, and we expect $a_{ret} = 0$, $a_{eye} > 0$; by contrast, in the R+T geometry, $p' = 1$, and we expect $a_{ret} > 0$, $a_{eye} = 0$. To test this prediction, this linear model was fit to the direction reports in each of the conditions by minimizing the mean cosine error with L1 regularization to impose sparsity:

$$\text{argmin}_{a_{ret}, a_{eye}} \frac{1}{N}\sum_{i=1}^{N}(1 - \cos(\Theta(\widehat{\omega}_i) - \Theta(\omega_i))) + \alpha(|a_{ret}| + |a_{eye}|). \tag{13}$$

Here, $\Theta(\widehat{\omega}_i)$ and $\Theta(\omega_i)$ indicate the predicted and actual reported object motion directions in the $i$-th trial. Regularization strength, $\alpha$, was chosen by cross-validation. Optimization was done using the *fminsearch* function in MATLAB (Mathworks, MA). $a_{ret}$ and $a_{eye}$ were bounded in the range of [0,1].

For Experiment 2, a cumulative-Gaussian psychometric function was fit to binary depth reports in each viewing geometry using the *psignifit* library[121] in MATLAB:

$$\psi(x; m, w, \lambda, \gamma) = \gamma + (1 - \lambda - \gamma)S(x; m, w) \tag{14}$$

where $\lambda$ and $\gamma$ denote the lapse rate at the highest and lowest stimulus levels. $S$ is the cumulative Gaussian function:

$$S(x; m, w) = \Phi\left(C\frac{x - m}{w}\right) \tag{15}$$

where $x$ is the retinal direction of the object, $m$ and $w$ are the mean and standard deviation of the Gaussian function, respectively, and $C = \Phi^{-1}(0.95) - \Phi^{-1}(0.05)$. Confidence intervals around parameters were obtained by bootstrapping inference provided in the *psignifit* library[121].

**Recurrent neural networks and neural data**

**Architecture.** RNN models were implemented using the PsychRNN library[122] and Tensorflow[123]. The RNN consisted of three input units, 64 recurrent units, and two output units. Our goal is to model the inputs and outputs relevant to the psychophysical experiment while keeping the network structure simple. For inputs, we use one input unit to represent the horizontal component of the object's retinal motion, and the other two units to represent the horizontal components of background optic flow for two different depths. We reasoned that the minimum information needed to disambiguate the viewing geometry (R vs. R+T) is the flow vector of two background dots, one at a near

depth and the other at a far depth. The two outputs of the network were scalars representing the horizontal component of the object's motion in the world and its depth (positive means far and negative means near). The vertical component of the object's motion was not incorporated into the network because our experiments involved horizontal eye movements, which would interact only with the horizontal component of image motion. However, training the network to produce both horizontal and vertical motion yielded similar results. Recurrent units were fully connected; each unit received all inputs and was connected to both outputs. The dynamics of the network can be described as:

$$\tau d\mathbf{x} = (-\mathbf{x} + \mathbf{W}_{rec}\boldsymbol{r} + \mathbf{b}_{rec} + \mathbf{W}_{in}\mathbf{u})dt \qquad (16)$$

$$\mathbf{r} = \tanh(\mathbf{x}) \qquad (17)$$

$$\mathbf{z} = \mathbf{W}_{out}\boldsymbol{r} + \mathbf{b}_{out} \qquad (18)$$

where $\mathbf{u}$, $\mathbf{r}$, $\mathbf{x}$, and $\mathbf{z}$ denote the input, activation of the hidden layer, recurrent state, and output. $\tau$ and $dt$ are the predefined time constant and time step, respectively, and $\tau = 100$ ms, $dt = 10$ ms. $\mathbf{W}_{in}$, $\mathbf{W}_{rec}$, and $\mathbf{W}_{out}$ are the learnable weight matrices for input, recurrent, and output connections. $\mathbf{b}_{rec}$ and $\mathbf{b}_{out}$ are biases fed into the recurrent and output units.

**Task and training.** In the initial 500-ms period of each simulated trial, the values of the inputs were independent Gaussian noise, $\mathcal{N}(0, 0.5)$, representing sensory noise. After that, the stimulus was presented for 1 s, represented by a constant value of retinal motion, the horizontal component of the optic flow vector of a near dot, and the horizontal component of the optic flow of a far dot, in addition to the Gaussian noise. The scale of Gaussian noise was chosen to qualitatively match the slopes of the model's psychometric curves (Fig. 8B) to those observed in human participants (Supplementary Fig. S4). The stimulus presentation period was followed by another 500 ms of noise. The output channels corresponded to the horizontal velocity of the object in world coordinates and depth from MP, and the network was trained to minimize the total L2 loss on these outputs only during the last 500 ms of each trial, after stimulus presentation was completed. Optimization was done with the ADAM optimizer[124] implemented in TensorFlow[123]. There were 50,000 training epochs, and the learning rate was $1 \times 10^{-3}$. The batch size was 128. In each trial, the retinal and eye velocities were uniformly sampled from a range of −10 to 10 (arbitrary units).

**Psychometric functions of the RNN.** After training, psychometric functions for the motion estimation and depth discrimination tasks were obtained by running predictions of the RNN on a set of inputs that replicated the human psychophysical experiments. Retinal motion directions ranged from −90° to 90° with a spacing of 12°, and the speed was constant at 2 (arbitrary units). The speed of eye velocity was 3 times that of the retinal motion, and the directions were leftward and rightward. Horizontal components of the retinal motion and eye velocity were used as inputs, and the model's estimated object motion direction was obtained by taking the arctangent between the veridical vertical component of the object's motion and the model's estimate of its horizontal speed.

**Tuning of single units in the network.** After training, we tested the RNN on a grid of stimuli covering all retinal and eye velocity combinations ranging from −10 to 10 with a spacing of 1 (arbitrary units) and both viewing geometries. For each recurrent unit, the joint velocity tuning profile at each time point was obtained by mapping the activation of the test stimuli to the 2D velocity grid.

**Tuning shifts in recurrent units.** We quantified the extent of tuning shifts in each joint tuning profile as the degree of asymmetry in its 2D Fourier transform. A shift of retinal velocity tuning with eye velocity would manifest as a diagonal structure in the joint tuning profile[57], and such diagonal structures will produce an asymmetric 2D Fourier power spectrum[125]. Specifically, we took the 2D Fourier transform of the joint tuning profile at the last time point for each recurrent unit of the network and thresholded the power spectrum at −10 dB to reduce noise. We then computed the normalized product of inertia of the power spectrum as

$$I_{xy} = \frac{\sum_x \sum_y xy P(x,y)}{\sum_x \sum_y |xy| P(x,y)} \times 100 \qquad (19)$$

where $x$ and $y$ are coordinates in the 2D Fourier domain, and $P(x, y)$ is the power at $(x, y)$. The normalized product of inertia ranges from −100% to 100%, with 0% indicating no tuning shift, 100% being maximally shifted towards world coordinates, and −100% showing maximum tuning shifts in the opposite direction of world coordinates. This metric allows us to quantify the extent of tuning shifts without assuming a specific form of the joint tuning profile; therefore, it is more generally applicable than our previous measure using parametric model fitting[57].

**Tuning shifts in MT neurons.** Due to the limited samples in the 2D velocity space of the experimental data in ref. 57, we could not use the normalized product of inertia to measure tuning shifts in the neural responses of MT neurons. Instead, we used the estimated weights on eye velocity developed to measure the tuning shifts in MT neurons[57]. In brief, we modeled neural responses to eye velocity and retinal motion as a combination of tuning shift, multiplicative gain, and additive modulation:

$$\lambda = A[g(v_{eye})f(v_{retina} + w v_{eye}) + o(v_{eye})]^+ + B \qquad (20)$$

$$g(v) = \frac{2}{1 + \exp(-\alpha v)} \qquad (21)$$

$$f(v) = \exp\left(-\frac{1}{2\sigma^2}\left(\log\frac{|v| + \delta}{s + \delta}\right)^2\right) \exp \kappa(\cos(\Theta(v) - \varphi) - 1) \qquad (22)$$

$$o(v) = \frac{2}{1 + \exp(-\beta v)} - 1 \qquad (23)$$

Here, $\lambda$ is the estimated firing rate; $v_{retina}$ and $v_{eye}$ are retinal and eye velocities at each time point; $A$ and $B$ are the amplitude and baseline firing rate; $w$ is the weight on eye velocity that quantifies the extent of tuning shifts; $[\cdot]^+$ is a rectifier that prevents negative firing rates; $g(v)$ is the multiplicative gain function; $\alpha$ controls the slope of the gain function; $f(v)$ is the tuning function; $\sigma$, $\delta$, $s$, $\kappa$, and $\varphi$ jointly define the width and offset of the function; $|v|$ and $\Theta(v)$ denote the speed and direction of the velocity; $o(v)$ is the additive modulation function, and $\beta$ controls the slope of the additive function. Free parameters in the model were estimated by minimizing the negative log-likelihood assuming Poisson noise.

The estimated weights on eye velocity, $\omega$, range from −1 to 1, with 0 being retinal-centered, 1 being completely world-centered, and −1 being the opposite of the expected shift. While strictly speaking, this measure is not equivalent to the normalized product of inertia used for hidden units, they are bounded in the same range and are roughly linearly related. Therefore, we used them as measures of tuning shifts and compared the distributions of these metrics between RNN units and neurons in MT.

## Reporting summary

Further information on research design is available in the Nature Portfolio Reporting Summary linked to this article.

## Data availability

All data are available at https://doi.org/10.17605/OSF.IO/ZY8W6.

## Code availability

Code is available at https://doi.org/10.17605/OSF.IO/ZY8W6.

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

## Acknowledgements

This work was supported by National Institutes of Health grants U19NS118246 and R01EY013644 to GCD.

## Author contributions

Conceptualization: Z.X. and G.C.D. Methodology: Z.X., A.A., and G.C.D. Software: Z.X., A.A., and G.C.D. Formal analysis: Z.X. Investigation: Z.X. and J.P. Resources: A.A. and G.C.D. Data curation: Z.X. and J.P. Writing—original draft: Z.X. and G.C.D. Writing—review and editing: Z.X., J.P., A.A., and G.C.D. Visualization: Z.X., A.A., and G.C.D. Supervision: G.C.D. Funding acquisition: G.C.D.

## Competing interests

The authors declare no competing interests.
