## [Transparent Peer Review file · Nature Communications]

Flexible computation of object motion and depth based on viewing geometry inferred from optic flow

Corresponding Author: Dr Zhe-Xin Xu

Version 0:

Reviewer comments:

Reviewer #1

(Remarks to the Author)

In this original and interesting study, the authors propose that the visual optic flow is naturally and flexibly combined with eye movement-related cues, and used by human subjects to infer the viewing geometry of 3D visual motion. The article is well written and nicely illustrated (with example videos available online) and it proposed a novel theoretical framework which suggests that the sensory consequences of eye movements, rather than being a nuisance, are efficiently integrated to disambiguate the 3D structure of the visual scene. The complexity of the computations underlying the geometric model of visual stimuli is explained in appreciable depth (although I would personally like some more guidance, see below). Yet, given the difficult exercise that is required to follow the hypothesized motion estimation based on the 3D dynamics of multiple objects, I encourage the authors to provide more specific links between the known perceptual phenomena and the model's predictions (see below).

Major points:

Clarifications

The proposed theoretical framework seems solid and it encompasses several established notions concerning motion and depth perception, as well as eye movements. Yet, some statements are not as clear, or they are somehow "diluted" throughout the manuscript. I encourage the authors to provide additional clarifications (or point to those clarifications if I have missed them)

How does the proposed theoretical framework account for

- 1) The reduced sensitivity to visual motion coherent with the sensory consequence of smooth motion tracking (I was personally confused by the horizontal-left direction of the perceptual bias observed and modeled in the R condition for real and simulated pursuit to the right – I might have misunderstood the condition transformation)
- 2) The fact that smooth pursuit of a moving stimulus over a (static or) moving background leads to an enhanced or reduced pursuit gain depending on the coherence between target and background motion
- 3) the added value of eye movements related information (proprioception/efferent copy/predictive information)?
- 4) More in general, it would be helpful to explain how the new framework could explain the mentioned visual illusions during tracking eye movements (Filehne, Aubert-Fleischl...)
- 5) To make the questioning even more complex, it has been shown under many conditions that human participants track perceived rather than physical (retinal, single object) motion. This fact could induce some important 2nd order effect of recurrent signals to be taken into account in the proposed model inferring the viewing geometry based on retinal and eye motion properties. The authors might want to discuss this possibility

Speculation about implications and perspectives:

In the Discussion, it is said: "...our study demonstrates that humans utilize the visual consequences of smooth pursuit eye movements (i.e., optic flow) to infer their viewing geometry and adaptively compute the depth and motion of objects. » This is a very interesting point. Several studies point out that eye movements, in particular orienting saccades, microsaccades and small fixational eye movements, do serve the purpose of disambiguating the visual input and the can do that in a flexible, adaptive way (in particular related to the natural statistics of visual scenes). The present work suggests a generalization of this idea to the 3D interpretation of the world : should we deduce that « exploratory » eye and head movements could help 3D disambiguated perception (and navigation) under poor visibility (lack of other 3D cues)? Should we expect to observe an

optimal pattern of «3D active vision » involving head and eye movements?

Minor points

In the Methods : «failure to maintain fixation within a $\pm 5^\circ$ rectangular window around the fixation target resulted in a failed trial » this tolerance window is much larger than the ones commonly used to assess precise fixation : is there a reason for that ?

Stimuli luminance is also exceptionally low (I would say by a factor of 10 compared to standard stimuli used in motion tracking experiments)

The order of presentation of exp 1 and 2 (if not randomized, as it seems) could have biased the depth perception results (also, non-naive subjects should be avoided in this kind of paradigms...)

Supplementary material,

- Additional participants: it is not clear to me why these participants are « special » and why their data have not been analyzed and presented with the others ?

-details about stimulus generation: I actually appreciate to have some advanced details about this technical part, however a little more guidance would help the non-expert reader understand better: what is the 4th coordinate in the « 3D » coordinate vector X ?

Reviewer #2

(Remarks to the Author)

Summary:

In this article, the authors study how viewing geometry, informed by the optical flow of the scene context, changes how retinal motion is used to compute object depth and motion. To this end, they conducted psychophysical experiments that revealed a distinct pattern in the perceived direction of motion based on the implied geometry. They also investigated the underlying neural mechanism by comparing units in an RNN trained to perform the task to the tuning properties of MT neurons.

Overall, I find the theoretical framework and behavioral experiments to be well executed, and I particularly appreciate the tight comparisons between the theory and data. I also think the authors reported an interesting and novel phenomenon on how other motion cues in the scene change the motion perception of a specific object through the inferred viewing geometry.

I do have some questions regarding the interpretation of the (individual) variability observed in the data, which the authors should unpack in greater depth. Additionally, I found the RNN comparison part of the study to be somewhat speculative, and would like the authors to clarify further. I will detail my concerns below.

Major comments:

1. The example subject (h500) shown in Figure 6D is not representative of the overall pattern of the R + T data in the pursuit condition. Based on Figure S1, it appears that most subjects show an overall negative directional bias, instead of the vertical bias as predicted by the theory. The authors state that “this might indicate that these subjects interpreted the viewing geometry as a mixture between R and R + T”, but this explanation seems post-hoc, given that the R + T geometry is general and already contains the R geometry (i.e., with no translational velocity). Could the authors clarify their explanation and, ideally, provide a quantitative prediction of this interpretation in Figure 6C?

2. Related to the point above, should one expect a stronger correspondence between the measured motion and depth perception? For example, as shown in Figure 7D-E, subject h507 showed very similar depth perception of the object in the R + T configuration in both the Pursuit and Fixation conditions, which implies that the inferred viewing geometry is similar between the two. However, the reported motion direction is quite different in this case for subject h507 (Figure S1, C vs. G). I suspect this might have something to do with whether the eye rotation is self-generated, but I would like the authors to expand on this point more precisely/quantitatively based on their theory, especially interpreting the connection between the motion and depth experiment.

3. In Section 3.4, the authors demonstrated that neurons in an RNN trained to perform similar tasks exhibit a shift in speed tuning, from a more independent coding of retinal and eye velocity in the R geometry to joint coding in the R + T geometry. They also showed a qualitative similarity of this shift to MT neuron responses reported in a previous study. I am not entirely sure what conclusion should be drawn here. The similarity seems too weak and narrow to claim that MT neurons are performing the same computation as the trained RNN. The authors also did not appear to provide further analysis of the underlying mechanisms, such as how tuning properties change depending on the inferred viewing geometry, or why such shifts in tuning properties are necessary to support the flexible computation of motion based on geometry. Note that I am not suggesting that all of these analyses are necessary, but rather to point out that the overall message of this section is somewhat vague and speculative.

Minor comments:

1. Could the authors clarify how the angular velocities in Figure 3 are defined? For example, for the world velocity of a moving object, what is the reference frame used for angular velocity when the eye is both rotating and translating?
2. I find the organization of Figure 5 somewhat confusing. It may be more intuitive to include visual cues to indicate that the upper and lower parts of panel B correspond to and explain the respective sections of panel A, for example.
3. Line 317: Should there be two p-values, one for the pursuit condition, and one for the fixation condition?
4. In the RNN implementation, the network estimates only the horizontal velocity component, while the vertical velocity is provided directly. This simplification seems unnecessary, as it should be straightforward to implement the network to estimate both the horizontal and vertical components.

Version 1:

Reviewer comments:

Reviewer #1

(Remarks to the Author)

The authors have thoroughly answered to my questions and taken into account a few comments/suggestions of mine. I do not have other comments.

Reviewer #2

(Remarks to the Author)

I would like to thank the authors for their detailed response. Most of my previous comments have been properly addressed.

However, I would like to further clarify comment #2. More specifically, I am asking whether the a_{ret} parameter should also correlate with the depth perception (e.g., threshold) measured in the second experiment. I understand that a non-zero a_{eye} could change the measured motion direction, but the depth direction should be directly related to a_{ret} , since the depth cue in this case corresponds to the part of the motion interpreted as motion parallax, as the authors explained.

Version 2:

Reviewer comments:

Reviewer #2

(Remarks to the Author)

Thanks for the clarification. All my questions are now addressed by the authors.

Reviewer #1 (Remarks to the Author):

In this original and interesting study, the authors propose that the visual optic flow is naturally and flexibly combined with eye movement-related cues, and used by human subjects to infer the viewing geometry of 3D visual motion. The article is well written and nicely illustrated (with example videos available online) and it proposed a novel theoretical framework which suggests that the sensory consequences of eye movements, rather than being a nuisance, are efficiently integrated to disambiguate the 3D structure of the visual scene. The complexity of the computations underlying the geometric model of visual stimuli is explained in appreciable depth (although I would personally like some more guidance, see below).

We thank the reviewer for their supportive comments and for appreciating the novelty in our study.

Yet, given the difficult exercise that is required to follow the hypothesized motion estimation based on the 3D dynamics of multiple objects, I encourage the authors to provide more specific links between the known perceptual phenomena and the model's predictions (see below).

Major points:

Clarifications

The proposed theoretical framework seems solid and it encompasses several established notions concerning motion and depth perception, as well as eye movements. Yet, some statements are not as clear, or they are somehow "diluted" throughout the manuscript. I encourage the authors to provide additional clarifications (or point to those clarifications if I have missed them)

We appreciate the reviewer's requests to provide some additional connections to previous observations and literature. And we have done our best to address each of these points. Unfortunately, since the reviewer did not give specific pointers to the literature that they are referring to, we might not have correctly inferred the studies that they mention.

How does the proposed theoretical framework account for

1) The reduced sensitivity to visual motion coherent with the sensory consequence of smooth motion tracking

We think that the reviewer is referring to previous work (e.g., Bedell and Lott, 1996, Current Biology) that showed a reduced motion detection sensitivity when objects move

in the same direction as the optic flow resulting from smooth pursuit. This phenomenon is consistent with the R geometry in our framework. In the R geometry, smooth pursuit eye movement produces optic flow vectors opposite to eye velocity. Eq. 3 predicts that the observer perceives object motion in world coordinates by adding retinal and eye velocities, $v_{obj} = v_{ret} + v_{eye}$, which is equivalent to subtracting the optic flow from retinal motion. Therefore, as the retinal motion of the object becomes more similar to the optic flow, the perceived motion approaches zero, resulting in reduced sensitivity. We have added text to discuss this in lines 471-473.

(I was personally confused by the horizontal-left direction of the perceptual bias observed and modeled in the R condition for real and simulated pursuit to the right – I might have misunderstood the condition transformation)

It is sensible for the reviewer to have been confused by this, as the predictions in Fig. 6C show a leftward direction bias in the R geometry, while the schematics of Figure 5 illustrate a rightward shift. We apologize for this confusion, which was caused by some folding steps in our data visualization that were not explained adequately. Real and simulated pursuit to the right does indeed produce a *rightward* bias in the R condition, as the reviewer expected. In Figure 6C-E, we flipped the data for rightward pursuit to merge with those for leftward pursuit, therefore, it appears as if only leftward pursuit data was shown. To avoid confusion, the data folding and flipping steps are now illustrated in detail in Supplementary Figure S1, and we have added text to the caption of Figure 6 and lines 816-823 for further clarification.

2) The fact that smooth pursuit of a moving stimulus over a (static or) moving background leads to an enhanced or reduced pursuit gain depending on the coherence between target and background motion

We presume that the reviewer is referring to effects such as those summarized in Table 1 of the review by Spering and Gegenfurtner (2008, Brain Research). We conducted additional analysis on the pursuit gains and we indeed found a significantly greater pursuit gain in the R+T geometry, for which the background elements moved in the same direction as the pursuit target, as compared to the R geometry, for which all background elements remained stationary during pursuit. We now describe these observations in the text, lines 805-809, and in Table 1.

Because we did not find any significant correlations between pursuit gains and parameters from our model fitting, as stated in lines 809-813, we believe that this difference in pursuit gain does not alter the conclusions of the study.

3) the added value of eye movements related information (proprioception/efferent copy/predictive information)?

We are not certain exactly which types of effects the reviewer is referring to here, but we certainly agree with the reviewer that extra-retinal signals related to eye movements provide important additional information for visual processing that may not be substituted by optic flow. We had not intended to imply otherwise, and have now made this point explicit in lines 546-549. Our experiments were not specifically designed to quantify the relative contributions of optic flow and extra-retinal signals in inferring viewing geometry, and our model does not distinguish between the sources of eye velocity. We have now addressed this issue in the Discussion in lines 546-549, and given some examples of phenomena for which extra-retinal signals may be crucial.

4) More in general, it would be helpful to explain how the new framework could explain the mentioned visual illusions during tracking eye movements (Filehne, Aubert-Fleischl...)

The Filehne illusion and the Aubert-Fleischl paradox can be explained by a gain smaller than 1 in our R geometry, which indicates an underestimation of eye velocity. Thus, the conventional explanation for these phenomena is subsumed within our framework. We have now included this point in the Discussion section (lines 465-471). Below is a more in-depth explanation for the reviewer:

In the Filehne illusion, when the eye rotates in one direction, a stationary object is perceived to move in the opposite direction of the eye movement. This scenario is consistent with the R viewing geometry. According to Eq. 10, the perceived object motion $v_{obj} = (1 - a_{ret})v_{ret} + a_{eye}v_{eye}$. Because we expect $a_{ret} = 0$ in R geometry and $v_{ret} = -v_{eye}$ for stationary objects, we have $v_{obj} = v_{ret} + a_{eye}v_{eye} = (a_{eye} - 1)v_{eye}$. If the observer accounts for eye movement completely, $a_{eye} = 1$, then a stationary object is perceived: $v_{obj} = (1 - 1) \times v_{eye} = 0$. If eye movement is not fully accounted for, such as $a_{eye} = 0.6$, then $v_{obj} = (0.6 - 1) \times v_{eye} = -0.4 \times v_{eye}$, which is in the opposite direction of eye velocity.

In the Aubert-Fleischl phenomenon, a moving object is perceived as moving more slowly when smooth pursuit tracks that object. This is also consistent with the R geometry, and assuming perfect tracking, $v_{ret} = 0$. Eq. 10 gives: $v_{obj} = 0 + a_{eye}v_{eye}$. If eye movement is fully accounted for, $a_{eye} = 1$ and $v_{obj} = v_{eye}$, which is the correct answer. If eye movement is underestimated, $a_{eye} < 1$ and $v_{obj} < v_{eye}$, suggesting an underestimation of object motion during pursuit.

5) To make the questioning even more complex, it has been shown under many conditions that human participants track perceived rather than physical (retinal, single object) motion. This fact could induce some important 2nd order effect of recurrent signals to be taken into account in the proposed model inferring the viewing geometry based on retinal and eye motion properties. The authors might want to discuss this possibility

We appreciate the reviewer for pointing out these phenomena. Our current model only considers actual retinal motion of a target object and how it interacts with optic flow and smooth pursuit eye movements. How these second-order effects might be incorporated into our framework remains an interesting question for future studies, but it is not something that we can address in detail at this time. We have now included this limitation of our framework in the Discussion section (third paragraph under “Limitations and future directions”, lines 639-643), along with a couple of examples of the types of phenomena that we think the reviewer is mentioning.

Speculation about implications and perspectives:

In the Discussion, it is said: “...our study demonstrates that humans utilize the visual consequences of smooth pursuit eye movements (i.e., optic flow) to infer their viewing geometry and adaptively compute the depth and motion of objects. » This is a very interesting point. Several studies point out that eye movements, in particular orienting saccades, microsaccades and small fixational eye movements, do serve the purpose of disambiguating the visual input and they can do that in a flexible, adaptive way (in particular related to the natural statistics of visual scenes). The present work suggests a generalization of this idea to the 3D interpretation of the world : should we deduce that « exploratory » eye and head movements could help 3D disambiguated perception (and navigation) under poor visibility (lack of other 3D cues)? Should we expect to observe an optimal pattern of «3D active vision » involving head and eye movements?

We thank the reviewer for raising these questions, as we do think they are important to address. In our framework, the information about viewing geometry provided by optic flow only depends on how the eye moves relative to the scene, which could be a result of whole body movement, head movement, eye-in-socket rotation, or combinations of these movements. There is evidence suggesting that humans and other species use exploratory eye and head movements to produce optic flow for discerning the 3D layout of the environment, and we now provide some examples in the Discussion. We see no reason why the optic flow generated by these exploratory movements could not be used to infer viewing geometry. That said, our framework currently makes no assumptions or predictions for how eye, head, and body movements should be coordinated to generate a particular movement of the eye relative to the scene; that is well beyond the scope of

this study (though we agree that it is very interesting to explore). We now discuss these points in the Discussion section (lines 474-485).

Minor points

In the Methods : «failure to maintain fixation within a $\pm 5^\circ$ rectangular window around the fixation target resulted in a failed trial » this tolerance window is much larger than the ones commonly used to assess precise fixation : is there a reason for that ?

In both experiments, fixation and pursuit trials were randomly intermixed within each session, and a $\pm 5^\circ$ window is not uncommon in studies of smooth pursuit. For example, a window of $\pm 3^\circ$ - 5° was used in Hohl, Chaisanguanthum, and Lisberger (2013) and $\pm 3^\circ$ - 4° was used in Stone and Lisberger (1990).

Stimuli luminance is also exceptionally low (I would say by a factor of 10 compared to standard stimuli used in motion tracking experiments)

We have reported the mean luminance of the aperture that contained the object (including dots and the dark background within the aperture), not the brightness of the individual dots on the display. This is why the mean luminance values appeared to be low. We have now clarified this in the Apparatus section (lines 674-676).

The order of presentation of exp 1 and 2 (if not randomized, as it seems) could have biased the depth perception results (also, non-naive subjects should be avoided in this kind of paradigms...)

We thank the reviewer for raising this point, as there are a couple of valuable clarifications to make. All participants completed Experiment 2 before Experiment 1, which we had previously failed to mention. Therefore, performance in the depth task in Experiment 2 could not have been biased by subjects performing Experiment 1. We have now included information about the experiment order in the Methods section (lines 660-661). Although the task order was fixed, we think that performing the binary depth discrimination task in Experiment 2 would be unlikely to affect the results of the analog motion estimation task in Experiment 1.

We had previously labeled data from non-naive participants in lighter shades in the original version of Figure 6G-H, but we failed to mention this distinction in the figure caption. We have now emphasized these data by using dashed lines for the non-naive subjects and we have also provided a list of naive participants in the Methods section (line 658). Because we see no consistent differences between the patterns of results

shown by naive and non-naive participants, we think it is valuable to include all of the data. Among the 3 subjects whose data conform least well to our predictions, we now also note (line 290-291) that one of these 3 is a non-naive subject.

Supplementary material,

- Additional participants: it is not clear to me why these participants are « special » and why their data have not been analyzed and presented with the others ?

This was clearly a misunderstanding, and we apologize for the confusion. These participants are in addition to the example subjects shown in Figures 6D-E and 7D-E. They are not special in any way, and the data from all subjects are included in the summaries shown in Figures 6G-H and 7G-H. We have now modified the captions of Supplementary Figures S3 and S4 to make this clear. The goal of S3 and S4 is simply to show the raw data from all subjects, rather than only showing example subjects.

-details about stimulus generation: I actually appreciate to have some advanced details about this technical part, however a little more guidance would help the non-expert reader understand better: what is the 4th coordinate in the « 3D » coordinate vector X ?

We have now included an explanation for this on the Supplementary Information page ix: “Homogeneous coordinates were used so that 3D projection, translation, and rotation can be conveniently written as matrix multiplications (Bloomenthal and Rokne, 1994). The fourth dimension, w , allowed us to represent points at infinity easily as $w = 0$. When $w = 1$, homogeneous coordinates would be the same as 3D Cartesian coordinates. This conversion is conventional in computer graphics including the OpenGL library we used (Woo et al., 1999; Bloomenthal and Rokne, 1994).”

Reviewer #2 (Remarks to the Author):

Summary:

In this article, the authors study how viewing geometry, informed by the optical flow of the scene context, changes how retinal motion is used to compute object depth and motion. To this end, they conducted psychophysical experiments that revealed a distinct pattern in the perceived direction of motion based on the implied geometry. They also investigated the underlying neural mechanism by comparing units in an RNN trained to perform the task to the tuning properties of MT neurons.

Overall, I find the theoretical framework and behavioral experiments to be well executed, and I particularly appreciate the tight comparisons between the theory and data. I also think the authors reported an interesting and novel phenomenon on how other motion cues in the scene change the motion perception of a specific object through the inferred viewing geometry.

We thank the review for their positive comments.

I do have some questions regarding the interpretation of the (individual) variability observed in the data, which the authors should unpack in greater depth. Additionally, I found the RNN comparison part of the study to be somewhat speculative, and would like the authors to clarify further. I will detail my concerns below.

Major comments:

1. The example subject (h500) shown in Figure 6D is not representative of the overall pattern of the R + T data in the pursuit condition. Based on Figure S1, it appears that most subjects show an overall negative directional bias, instead of the vertical bias as predicted by the theory. The authors state that “this might indicate that these subjects interpreted the viewing geometry as a mixture between R and R + T”, but this explanation seems post-hoc, given that the R + T geometry is general and already contains the R geometry (i.e., with no translational velocity). Could the authors clarify their explanation and, ideally, provide a quantitative prediction of this interpretation in Figure 6C?

We agree with the reviewer that the example subject originally shown in Figure 6D is not representative of the general pattern of results for R+T Pursuit condition. We have now changed the example subject to h201, which shows a more representative pattern of results. To address the reviewer’s concern about the interpretation of the downward shift of the blue curves in the Pursuit condition, we have now included a new Supplementary Figure 2 to illustrate how our model captures this negative directional bias in the R+T Pursuit condition. We now mention this new figure and how the model captures this effect in the text (lines 297-300). Finally, we would not say that the R+T geometry “contains the R geometry”, as the reviewer stated, because only in the R geometry is the optic flow depth invariant, and CT may be performed.

2. Related to the point above, should one expect a stronger correspondence between the measured motion and depth perception? For example, as shown in Figure 7D-E, subject h507 showed very similar depth perception of the object in the R + T configuration in both the Pursuit and Fixation conditions, which implies that the inferred viewing geometry is similar between the two. However, the reported motion direction is

quite different in this case for subject h507 (Figure S1, C vs. G). I suspect this might have something to do with whether the eye rotation is self-generated, but I would like the authors to expand on this point more precisely/quantitatively based on their theory, especially interpreting the connection between the motion and depth experiment.

We thank the reviewer for raising this excellent question. We have further examined our model predictions to assess whether there is a linkage between the negative direction bias pointed out by the review in the previous comment and performance of subjects in the depth discrimination task. Interestingly, we found that, while our model captures the negative directional bias in the R+T Pursuit condition of the motion task, this effect did not produce any corresponding differences in predicted performance of the depth discrimination task, as now illustrated in the new Supplementary Figure S2 and mentioned in the text (lines 302-304). We believe that this occurs because Experiment 2 only measured the perception of depth sign (a binary near vs. far choice), not a continuous estimate of depth magnitude.

3. In Section 3.4, the authors demonstrated that neurons in an RNN trained to perform similar tasks exhibit a shift in speed tuning, from a more independent coding of retinal and eye velocity in the R geometry to joint coding in the R + T geometry. They also showed a qualitative similarity of this shift to MT neuron responses reported in a previous study. I am not entirely sure what conclusion should be drawn here. The similarity seems too weak and narrow to claim that MT neurons are performing the same computation as the trained RNN. The authors also did not appear to provide further analysis of the underlying mechanisms, such as how tuning properties change depending on the inferred viewing geometry, or why such shifts in tuning properties are necessary to support the flexible computation of motion based on geometry. Note that I am not suggesting that all of these analyses are necessary, but rather to point out that the overall message of this section is somewhat vague and speculative.

The reviewer makes a fair point that the comparison between the RNN and the MT is incomplete, and thus speculative. We agree with the reviewer that this does not provide direct evidence supporting MT neurons as the neural basis for these flexible computations. Unfortunately, a more direct examination of this question requires a new neurophysiological study, in which the activity of MT neurons is recorded during presentation of the same types of stimuli used in this study. We are currently in the process of conducting these neurophysiological recordings to better understand the neural basis of these computations. That said, we still think the RNN comparison adds value to this manuscript. Thus, we have added text to the Discussion (lines 606-611) to clearly acknowledge the limitations of this comparison and to point out the need for new studies that directly compare responses of MT neurons under the same stimulus conditions used here.

Minor comments:

1. Could the authors clarify how the angular velocities in Figure 3 are defined? For example, for the world velocity of a moving object, what is the reference frame used for angular velocity when the eye is both rotating and translating?

Angular velocities of the eye and object are defined relative to a fixed point in the world, and retinal velocities are defined relative to the eye. We have now included this in the caption of Figure 3 and the Methods section (lines 831-833).

2. I find the organization of Figure 5 somewhat confusing. It may be more intuitive to include visual cues to indicate that the upper and lower parts of panel B correspond to and explain the respective sections of panel A, for example.

We thank the reviewer for pointing out this issue. We have added a horizontal line dividing the top and bottom panels of Figure 5, to aid visual clarity.

3. Line 317: Should there be two p-values, one for the pursuit condition, and one for the fixation condition?

For group-level statistics, we reported two sets of p-values, one for each condition, in lines 283-289. We have now added a new Supplementary Table S1 to document the statistics for each condition and each individual subject.

4. In the RNN implementation, the network estimates only the horizontal velocity component, while the vertical velocity is provided directly. This simplification seems unnecessary, as it should be straightforward to implement the network to estimate both the horizontal and vertical components.

This simplification was done so that both motion and depth tasks produce only one scalar estimate, which balances their contributions to the loss function used in training the network. We have tested a version of the network in which both horizontal and vertical motion components were incorporated, and the results remained similar after carefully balancing the contributions to the loss function for the two tasks. We prefer to maintain the simplified version since the vertical motion component was not manipulated or affected in our psychophysical experiments. We now comment on these issues in the Methods section (lines 873-877).

Reviewer #1 (Remarks to the Author):

The authors have thoroughly answered to my questions and taken into account a few comments/suggestions of mine. I do not have other comments.

Reviewer #2 (Remarks to the Author):

I would like to thank the authors for their detailed response. Most of my previous comments have been properly addressed.

However, I would like to further clarify comment #2. More specifically, I am asking whether the a_{ret} parameter should also correlate with the depth perception (e.g., threshold) measured in the second experiment. I understand that a non-zero a_{eye} could change the measured motion direction, but the depth direction should be directly related to a_{ret} , since the depth cue in this case corresponds to the part of the motion interpreted as motion parallax, as the authors explained.

We thank the reviewer for clarifying this comment. We have further examined this question and did not find any significant correlation between the individual a_{ret} estimates in Experiment 1 and the slopes measured in Experiment 2. While our theory predicts a relationship between a_{ret} and perceived depth magnitude, this relationship may not directly translate to the depth sign discriminability measured by Experiment 2. Although perceived depth magnitude could affect depth-sign judgments if depth magnitude estimates are strongly compressed toward the fixation plane, this relationship may not be strong and is likely to be obscured by other sources of individual variability. There is likely to be individual variability in depth-sign discrimination thresholds that cannot be accounted for by our linear model, such as variations in internal noise on representations of retinal or eye velocity. Further work that simultaneously measures both perceived motion direction and depth magnitude would allow for a more direct assessment of this relationship. We now discuss this issue in the Discussion section (lines 500-511).

REVIEWERS' COMMENTS

Reviewer #2 (Remarks to the Author):

Thanks for the clarification. All my questions are now addressed by the authors.

We thank the reviewers for their comments.